# Robust Optimal Transport with Applications in Generative Modeling and Domain Adaptation

**Yogesh Balaji**
Department of Computer Science
University of Maryland
College Park, MD
yogesh@umd.edu

**Rama Chellappa**
Electrical and Computer Engineering
and Biomedical Engineering Departments
Johns Hopkins University
Baltimore, MD
rchella4@jhu.edu

**Soheil Feizi**
Department of Computer Science
University of Maryland
College Park, MD
sfeizi@cs.umd.edu

## Abstract

Optimal Transport (OT) distances such as Wasserstein have been used in several areas such as GANs and domain adaptation. OT, however, is very sensitive to outliers (samples with large noise) in the data since in its objective function, every sample, including outliers, is weighed similarly due to the marginal constraints. To remedy this issue, robust formulations of OT with unbalanced marginal constraints have previously been proposed. However, employing these methods in deep learning problems such as GANs and domain adaptation is challenging due to the instability of their dual optimization solvers. In this paper, we resolve these issues by deriving a computationally-efficient dual form of the robust OT optimization that is amenable to modern deep learning applications. We demonstrate the effectiveness of our formulation in two applications of GANs and domain adaptation. Our approach can train state-of-the-art GAN models on noisy datasets corrupted with outlier distributions. In particular, the proposed optimization method computes weights for training samples reflecting how difficult it is for those samples to be generated in the model. In domain adaptation, our robust OT formulation leads to improved accuracy compared to the standard adversarial adaptation methods. Our code is available at https://github.com/yogeshbalaji/robustOT.

## 1 Introduction

Estimating distances between probability distributions lies at the heart of several problems in machine learning and statistics. A class of distance measures that has gained immense popularity in several machine learning applications is Optimal Transport (OT) [27]. In OT, the distance between two probability distributions is computed as the minimum cost of transporting a source distribution to the target distribution under some transportation cost function. Optimal transport enjoys several nice properties including structure preservation, existence in smooth and non-smooth settings, being well defined for discrete and continuous distributions [27], etc.

Two recent applications of OT in machine learning include generative modeling and domain adaptation. In Wasserstein GAN [1], a generative model is trained by minimizing the (approximate) Wasserstein distance between real and generative distributions. In the dual form, this objective

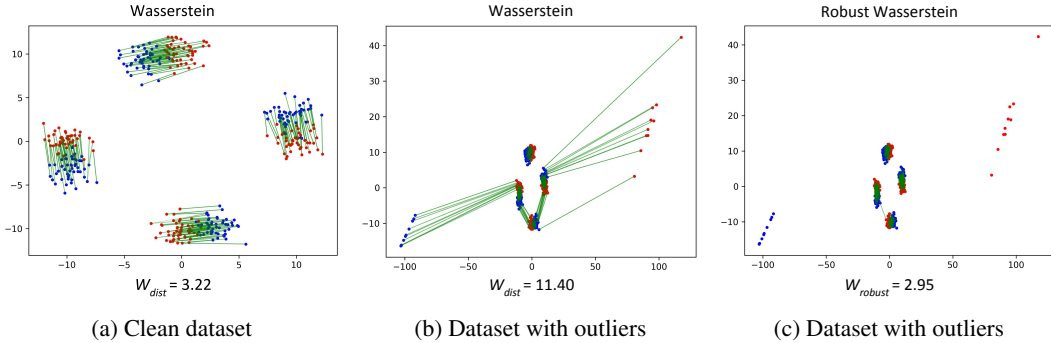

| | | |
|---|---|---|
| $W_{dist}$ = 3.22 | $W_{dist}$ = 11.40 | $W_{robust}$ = 2.95 |
| (a) Clean dataset | (b) Dataset with outliers | (c) Dataset with outliers |

Figure 1: Visualizing couplings of Wasserstein computation between two distributions shown in red and blue. In (a), we show the couplings when no outliers are present. In (b), we show the couplings when 5% outliers are added to the data. The Wasserstein distance increases significantly indicating high sensitivity to outliers. In (c), we show the couplings produced by the Robust Wasserstein measure. Our formulation effectively ignores the outliers yielding a Wasserstein estimate that closely approximates the true Wasserstein distance.

reduces to a two-player *min-max* game between a generator and a discriminator. Similar ideas involving distance minimization between source and target feature distributions are common in domain adaptation [10]. In [24, 2], Wasserstein distance is used as the choice of distance measure for minimizing the domain gap.

One of the fundamental shortcomings of optimal transport is its sensitivity to outlier samples. By outliers, we mean samples with large noise. In the OT optimization, to satisfy the marginal constraints between the two input distributions, every sample is weighed equally in the feasible transportation plans. Hence, even a few outlier samples can contribute significantly to the OT objective. This leads to poor estimation of distributional distances when outliers are present. An example is shown in Fig. 1, where the distances between distributions shown in red and blue are computed. In the absence of outliers (Fig. 1(a)), proper couplings (shown in green) are obtained. However, even in the presence of a very small fraction of outliers (as small as 5%), poor couplings arise leading to a large change in the distance estimate (Fig. 1(b)).

The OT sensitivity to outliers is undesirable, especially when we deal with large-scale datasets where the noise is inevitable. This sensitivity is a consequence of exactly satisfying the marginal constraints in OT's objective. Hence, to boost OT's robustness against outliers, we propose to utilize recent formulations of unbalanced optimal transport [6, 13] which relax OT's marginal constraints. The authors in [6, 13] provide an exact dual form for the unbalanced OT problem. However, we have found that using this dual optimization in large-scale deep learning applications such as GANs results in poor convergence and an unstable behaviour (see Section 3.1 and the appendix for details).

To remedy this issue, in this work, we derive a computationally efficient dual form for the unbalanced OT optimization that is suited for practical deep learning applications. Our dual simplifies to a weighted OT objective, with low weights assigned to outlier samples. These instance weights can also be useful in interpreting the difficulty of input samples for learning a given task. We develop two solvers for this dual problem based on either a discrete formulation or a continuous stochastic relaxation. These solvers demonstrate high stability in large-scale deep learning applications.

We show that, under mild assumptions, our robust OT measure (which is similar in form to the unbalanced OT) is upper bounded by a constant factor of the true OT distance (OT ignoring outliers) for any outlier distribution. Hence, our robust OT can be used for effectively handling outliers. This is visualized in Figure 1(c), where couplings obtained by robust OT effectively ignores outlier samples, yielding a good estimate of the true OT distance. We demonstrate the effectiveness of the proposed robust OT formulation in two large-scale deep learning applications of generative modeling and domain adaptation. In generative modeling, we show how robust Wasserstein GANs can be trained using state-of-the-art GAN architectures to effectively ignore outliers in the generative distrubution. In domain adaptation, we utilize the robust OT framework for the challenging task of synthetic to real adaptation, where our approach improves adversarial adaptation techniques by $\sim 5\%$.

## 2 Background on Optimal Transport

Let $\mathcal{X}$ denote a compact metric space, and $Prob(\mathcal{X})$ denote the space of probability measures defined on $\mathcal{X}$. Given two probability distributions $\mathbb{P}_X$, $\mathbb{P}_Y \in Prob(\mathcal{X})$ and a continuous cost function $c : \mathcal{X} \times \mathcal{X} \to \mathbb{R}$, optimal transport finds the minimum cost for transporting the density $\mathbb{P}_X$ to $\mathbb{P}_Y$ [27], given by the following cost function

$$\mathcal{W}(\mathbb{P}_X, \mathbb{P}_Y) = \min_{\pi \in \Pi(\mathbb{P}_X, \mathbb{P}_Y)} \int \int c(x, y)\pi(x, y) dx dy \tag{1}$$

where $\Pi(\mathbb{P}_X, \mathbb{P}_Y)$ is set of all joint distributions (couplings) whose marginals are $\mathbb{P}_X$ and $\mathbb{P}_Y$, respectively. That is, $\int \pi(x, y) dy = \mathbb{P}_X$ and $\int \pi(x, y) dx = \mathbb{P}_Y$. These set of constraints enforce that the marginals of couplings exactly match the distributions $\mathbb{P}_X$ and $\mathbb{P}_Y$. Hence, when the input distributions have outliers, they are forced to have non-zero weights in couplings leading to a large transportation cost. In practice, the dual form of the optimization (1) is often used.

**Kantrovich Duality:** The dual formulation of optimal transport problem [27] is given by:

$$\min_{\pi \in \Pi(\mathbb{P}_X, \mathbb{P}_Y)} \int \int c(x, y)\pi(x, y) dx dy = \max_{\phi \in Lip-1} \int \phi(x) d\mathbb{P}_X - \int \phi(x) d\mathbb{P}_Y. \tag{2}$$

Optimization (2) is the celebrated *Kantorovich-Rubinstein* duality. This is a simpler optimization problem compared to (1) since the maximization is over a class of 1-Lipschitz functions. In practice, $\phi(\cdot)$ function is implemented using a neural network, and the 1-Lipscitz constraint is enforced using weight clipping [1] or a penalty on the gradients [11].

The use of optimal transport has gained popularity in machine learning [26, 24, 1], computer vision [25, 4] and many other disciplines. Several relaxations of the OT problem have been proposed in the literature. Two popular ones include entropy regularization [8, 3] and marginal relaxation [13, 6, 7, 9, 12]. In this work, we utilize the marginal relaxations of [13, 6] for handling outlier noise in machine learning applications involving OT. To the best of our knowledge, ours is the first work to demonstrate the utility of unbalanced OT in large-scale deep learning applications. Only other paper that is similar in spirit to our work is [28]. However, [28] provides a relaxation for the Monge unbalanced OT, which is different from the unbalanced Kantrovich problem we consider in this paper.

## 3 Robust Optimal Transport

Our objective is to handle outliers in deep learning applications involving OT. For this, we use relaxed OT formulations. In this section, we first formally define the outlier model we use. Then, we discuss the existing marginal relaxation formulations in OT and the issues that arise in deep learning when using these formulations. We then propose a reformulation of the dual that is suited for deep learning.

**Outlier Model:** We consider outliers as samples with large noise. More specifically, let $\mathbb{P}_X$ and $\mathbb{P}_Y$ be two distributions whose Wasserstein distance we desire to compute. Let $\mathbb{P}_X = \alpha \mathbb{P}_X^c + (1 - \alpha)\mathbb{P}_X^a$; i.e., the clean distribution $\mathbb{P}_X^c$ is corrupted with $(1 - \alpha)$ fraction of noise $\mathbb{P}_X^a$. Then, $\mathbb{P}_X^a$ is considered an oulier distribution if $\mathcal{W}(\mathbb{P}_X^c, \mathbb{P}_Y) \ll \mathcal{W}(\mathbb{P}_X^a, \mathbb{P}_Y)$. For an example, refer to Fig. 1(b).

### 3.1 Unbalanced Optimal Transport

As seen in Fig. 1, sensitivity to outliers arises due to the marginal constraints in OT. If the marginal constraints are relaxed in a way that the transportation plan does not assign large weights to outliers, they can effectively be ignored. [6, 13] have proposed one such relaxation using $f$-divergence on marginal distributions. This formulation, called *Unbalanced Optimal Transport*, can be written as

$$\mathcal{W}^{ub}(\mathbb{P}_X, \mathbb{P}_Y) = \min_{\pi \in \Pi(\mathbb{P}_{\tilde{X}}, \mathbb{P}_{\tilde{Y}})} \int c(x, y)\pi(x, y) dx dy + \mathcal{D}_f(\mathbb{P}_{\tilde{X}} || \mathbb{P}_X) + \mathcal{D}_f(\mathbb{P}_{\tilde{Y}} || \mathbb{P}_Y) \tag{3}$$

where $\mathcal{D}_f$ is the $f$-divergence between distributions, defined as $\mathcal{D}_f(P || Q) = \int f(\frac{dP}{dQ}) dQ$. Furthermore, [13] derived a dual form for the problem. Let $f$ be a convex lower semi-continuous function.

Define $r^*(x) := \sup_{s>0} \frac{x-f(s)}{s}$ where $f'_\infty := \lim_{s\to\infty} \frac{f(s)}{s}$. Then,

$$\mathcal{W}^{ub}(\mathbb{P}_X, \mathbb{P}_Y) = \max_{\phi,\psi} \int \phi(x)d\mathbb{P}_X + \int \psi(y)d\mathbb{P}_Y \tag{4}$$
$$\text{s.t. } r^*(\phi(x)) + r^*(\psi(y)) \le c(x,y)$$

**Computational issues using this dual form in deep learning:**  Training neural networks using this dual form is challenging as it involves maximizing over *two* discriminator functions ($\phi$ and $\psi$), with constraints connecting these functions. For $\chi^2$ divergence, we derived the GAN objective using this dual and trained a model. However, we were unsuccessful in making the model converge using standard SGD as it showed severe instability. Please refer to Appendix for more details. This limits the utility of this formulation in deep learning applications. In what follows, we present a reformulation of the dual that is scalable and suited for deep learning applications.

### 3.2   Our Duality

We start with a slightly different form than (3) where we keep the $f$-divergence relaxations of marginal distributions as constraints:

$$\mathcal{W}^{rob}_{\rho_1,\rho_2}(\mathbb{P}_X, \mathbb{P}_Y) := \min_{\mathbb{P}_{\tilde{X}}, \mathbb{P}_{\tilde{Y}} \in Prob(\mathcal{X})} \min_{\pi \in \Pi(\mathbb{P}_{\tilde{X}}, \mathbb{P}_{\tilde{Y}})} \int\int c(x,y)\pi(x,y)dxdy \tag{5}$$
$$\text{s.t. } \mathcal{D}_f(\mathbb{P}_{\tilde{X}}||\mathbb{P}_X) \le \rho_1, \ \mathcal{D}_f(\mathbb{P}_{\tilde{Y}}||\mathbb{P}_Y) \le \rho_2.$$

In this formulation, we optimize over the couplings whose marginal constraints are the relaxed distributions $\mathbb{P}_{\tilde{X}}$ and $\mathbb{P}_{\tilde{Y}}$. To prevent over-relaxation of the marginals, we impose a constraint that the $f$-divergence between the relaxed and the true marginals are bounded by constants $\rho_1$ and $\rho_2$ for distributions $\mathbb{P}_{\tilde{X}}$ and $\mathbb{P}_{\tilde{Y}}$, respectively. As seen in Fig. 1(c), this relaxation effectively ignores the outlier distributions when $(\rho_1, \rho_2)$ are chosen appropriately. We study some properties of robust OT in the appendix. Notably, robust OT does not satisfy the triangle inequality.

Note that the Lagrangian relaxation of optimization (5) takes a similar form to that of the unbalanced OT objective (3). Having a hard constraint on $f$-divergence gives us an explicit control over the extent of the marginal relaxation which is suited for handling outliers. This subtle difference in how the constraints are imposed leads to a dual form of our robust OT that can be computed efficiently for deep learning applications compared to that of the unbalanced OT dual.

We consider the $\ell_2$ distance as our choice of cost function in the OT formulation. In this case, the OT distance is also called the *Wasserstein* distance. In that case, we have the following result:

**Theorem 1.** *Let $\mathbb{P}_X$ and $\mathbb{P}_Y$ be two distributions defined on a metric space. The robust Wasserstein measure admits the following dual form*

$$\mathcal{W}^{rob}_{\rho_1,\rho_2}(\mathbb{P}_X, \mathbb{P}_Y) = \min_{\mathbb{P}_{\tilde{X}}, \mathbb{P}_{\tilde{Y}}} \max_{D(.)\in Lip-1} \int D(x)d\mathbb{P}_{\tilde{X}} - \int D(x)d\mathbb{P}_{\tilde{Y}} \tag{6}$$
$$s.t \ \mathcal{D}_f(\mathbb{P}_{\tilde{X}}||\mathbb{P}_X) \le \rho_1, \ \mathcal{D}_f(\mathbb{P}_{\tilde{Y}}||\mathbb{P}_Y) \le \rho_2.$$

Thus, we can obtain a dual form for robust OT similar to the *Kantrovich-Rubinstein* duality. The key difference of this dual form compared to the unbalanced OT dual (opt. (4)) is that we optimize over a single dual function $D(.)$ as opposed to two dual functions in (4). This makes our formulation suited for deep learning applications such as GANs and domain adaptation. Note that the integrals in opt. (6) are taken with respect to the relaxed distributions $\mathbb{P}_{\tilde{X}}$ and $\mathbb{P}_{\tilde{Y}}$ which is a non-trivial computation.

In particular, we present two approaches for optimizing the dual problem (6):

**Discrete Formulation.**   In practice, we observe empirical distributions $\mathbb{P}_X^{(m)}$ and $\mathbb{P}_Y^{(n)}$ from the population distributions $\mathbb{P}_X$ and $\mathbb{P}_Y$, where $m$ and $n$ are sample sizes. Let $\{\mathbf{x}_i\}_{i=1}^m$, $\{\mathbf{y}_i\}_{i=1}^n$ be the samples corresponding to the empirical distribution $\mathbb{P}_X^{(m)}$ and $\mathbb{P}_Y^{(m)}$, respectively. Following [18], we use weighted empirical distribution for the perturbed distribution $\mathbb{P}_{\tilde{X}}$, i.e., $\mathbb{P}_X^{(m)}(\mathbf{x}_i) = 1/m$ and $\mathbb{P}_{\tilde{X}}(\mathbf{x}_i) = w_i^x$. Let $\mathbf{w}_x = [w_1^x, \ldots w_m^x]$. For $\mathbb{P}_{\tilde{X}}$ to be a valid pmf, $\mathbf{w}_x$ should lie in a simplex

($\mathbf{w}^x \in \Delta^m$) i.e., $w_i^x > 0$ and $\sum_i w_i^x = 1$. Then, the robust Wasserstein objective can be written as

$$\mathcal{W}_{\rho_1,\rho_2}^{rob}(\mathbb{P}_X,\mathbb{P}_Y) = \min_{\mathbf{w}_x \in \Delta^m, \mathbf{w}_y \in \Delta^n} \max_{\mathbf{D} \in Lip-1} (\mathbf{w}_x)^t \mathbf{d}_x - (\mathbf{w}_y)^t \mathbf{d}_y$$

$$\text{s.t} \quad \frac{1}{m}\sum_i f(mw_i^x) \le \rho_1, \quad \frac{1}{n}\sum_i f(nw_i^y) \le \rho_2$$

where $\mathbf{d}_x = [\mathbf{D}(\mathbf{x}_1), \mathbf{D}(\mathbf{x}_2)\dots\mathbf{D}(\mathbf{x}_m)]$, and $\mathbf{d}_y = [\mathbf{D}(\mathbf{y}_1), \mathbf{D}(\mathbf{y}_2)\dots\mathbf{D}(\mathbf{y}_n)]$. Since $f(.)$ is a convex function, the set of constraints involving $\mathbf{w}_x$ and $\mathbf{w}_y$ are convex w.r.t weights. We use $\chi^2$ as our choice of $f$-divergence for which $f(t) = (t-1)^2/2$. The optimization then becomes

$$\mathcal{W}_{\rho_1,\rho_2}^{rob}(\mathbb{P}_X,\mathbb{P}_Y) = \min_{\mathbf{w}_x \in \Delta^m, \mathbf{w}_y \in \Delta^n} \max_{\mathbf{D} \in Lip-1} (\mathbf{w}_x)^t \mathbf{d}_x - (\mathbf{w}_y)^t \mathbf{d}_y \tag{7}$$

$$\text{s.t} \quad \left\|\mathbf{w}_x - \frac{1}{m}\right\|_2 \le \sqrt{\frac{2\rho_1}{m}}, \quad \left\|\mathbf{w}_y - \frac{1}{n}\right\|_2 \le \sqrt{\frac{2\rho_2}{n}}$$

We solve this optimization using an alternating gradient descent between $\mathbf{w}$ and $\mathbf{D}$ updates. The above optimization is a second-order cone program with respect to weights $\mathbf{w}$ (for a fixed $\mathbf{D}$). For a fixed $\mathbf{w}$, $\mathbf{D}$ is optimized using stochastic gradient descent similar to [1].

**Continuous Stochastic Relaxation.** In (7), weight vectors $\mathbf{w}_x$ and $\mathbf{w}_y$ are optimized by solving a second order cone program. Since the dimension of weight vectors is the size of the entire dataset, solving this optimization is expensive for large datasets. Hence, we propose a continuous stochastic relaxation for (6). Let us assume that supports of $\mathbb{P}_{\tilde{X}}$ and $\mathbb{P}_X$ match (satisfied in real spaces). We make the following reparameterization: $\mathbb{P}_{\tilde{X}}(\mathbf{x}) = \mathbf{W}_x(\mathbf{x})\mathbb{P}_X(\mathbf{x})$. For $\mathbb{P}_{\tilde{X}}$ to be a valid pdf, we require $\int \mathbf{W}_x(\mathbf{x})d\mathbb{P}_X = 1$, i.e., $\mathbb{E}_{\mathbf{x}\sim\mathbb{P}_X}[\mathbf{W}_x(\mathbf{x})] = 1$. The constraint on $f$-divergence becomes $\mathbb{E}_{\mathbf{x}\sim\mathbb{P}_X}[f(\mathbf{W}_x(\mathbf{x}))] \le \rho_1$. Using these, the dual of robust Wasserstein measure can be written as

$$\mathcal{W}_{\rho_1,\rho_2}^{rob}(\mathbb{P}_X,\mathbb{P}_Y) = \min_{\mathbf{W}_x,\mathbf{W}_y} \max_{\mathbf{D}\in Lip-1} \mathbb{E}_{\mathbf{x}\sim\mathbb{P}_X}[\mathbf{W}_x(\mathbf{x})\mathbf{D}(\mathbf{x})] - \mathbb{E}_{\mathbf{y}\sim\mathbb{P}_Y}[\mathbf{W}_y(\mathbf{y})\mathbf{D}(\mathbf{y})] \tag{8}$$

$$\text{s.t} \quad \mathbb{E}_{\mathbf{x}\sim\mathbb{P}_X}[f(\mathbf{W}_x(\mathbf{x}))] \le \rho_1, \quad \mathbb{E}_{\mathbf{y}\sim\mathbb{P}_Y}[f(\mathbf{W}_y(\mathbf{y}))] \le \rho_2$$

$$\mathbb{E}_{\mathbf{x}\sim\mathbb{P}_X}[\mathbf{W}_x(\mathbf{x})] = 1, \quad \mathbb{E}_{\mathbf{y}\sim\mathbb{P}_Y}[\mathbf{W}_y(\mathbf{y})] = 1, \mathbf{W}_x(\mathbf{x}) \ge 0, \mathbf{W}_y(\mathbf{y}) \ge 0$$

$\mathbf{W}_x(.)$ and $\mathbf{W}_y(.)$ are weight functions which can be implemented using neural networks. One crucial benefit of the above formulation is that it can be easily trained using stochastic GD.

### 3.3 Can robust OT handle outliers?

**Theorem 2.** *Let $\mathbb{P}_X$ and $\mathbb{P}_Y$ be two distributions such that $\mathbb{P}_X$ is corrupted with $\gamma$ fraction of outliers i.e., $\mathbb{P}_X = (1-\gamma)\mathbb{P}_X^c + \gamma\mathbb{P}_X^a$, where $\mathbb{P}_X^c$ is the clean distribution and $\mathbb{P}_X^a$ is the outlier distribution. Let $\mathcal{W}(\mathbb{P}_X^a, \mathbb{P}_X^c) = k\mathcal{W}(\mathbb{P}_X^c, \mathbb{P}_Y)$, with $k \ge 1$. Then,*

$$\mathcal{W}_{\rho,0}^{rob}(\mathbb{P}_X,\mathbb{P}_Y) \le \max\left(1, 1 + k\gamma - k\sqrt{2\rho\gamma(1-\gamma)}\right)\mathcal{W}(\mathbb{P}_X^c, \mathbb{P}_Y).$$

The above theorem states that robust OT obtains a provably robust distance estimate under our outlier model. That is, the robust OT is upper bounded by a constant factor of the true Wasserstein distance. This constant depends on the hyper-parameter $\rho$: when $\rho$ is appropriately chosen, robust OT measure obtains a value approximately close to the true distance. Note that we derive this result for one-sided robust OT ($\mathcal{W}_{\rho,0}^{rob}$), which is the robust OT measure when marginals are relaxed only for one of the input distributions. This is the form we use for GANs and DA experiments (Section. 4).

**Choosing $\rho$ and the tightness of the bound:** The constant $\rho$ in our formulation is a hyper-parameter that needs to be estimated. The value of $\rho$ denotes the extent of marginal relaxation. In applications such as GANs or domain adaptation, performance on a validation set can be used for choosing $\rho$. Or when the outlier fraction $\gamma$ is known, an appropriate choice of $\rho$ is $\rho = \frac{\gamma}{2(1-\gamma)}$. More details and experiments on tightness of our upper bound are provided in the Appendix.

## 4 Experiments

For all our experiments, we use one-sided robust Wasserstein ($\mathcal{W}_{\rho,0}^{rob}$) where the marginals are relaxed only for one of the input distributions. Please refer to Appendix for all experimental details. Code for our experiments is available at `https://github.com/yogeshbalaji/robustOT`.

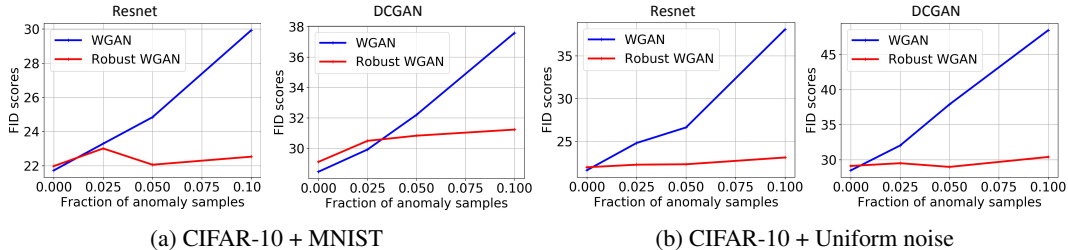

(a) CIFAR-10 + MNIST            (b) CIFAR-10 + Uniform noise

Figure 2: FID scores of GAN models trained on CIFAR-10 corrupted with outlier noise. In (a), samples from MNIST dataset are used as the outliers, while in (b), uniform noise is used. FID scores of WGAN increase with the increase in outlier fraction, while robust WGAN maintains FID scores.

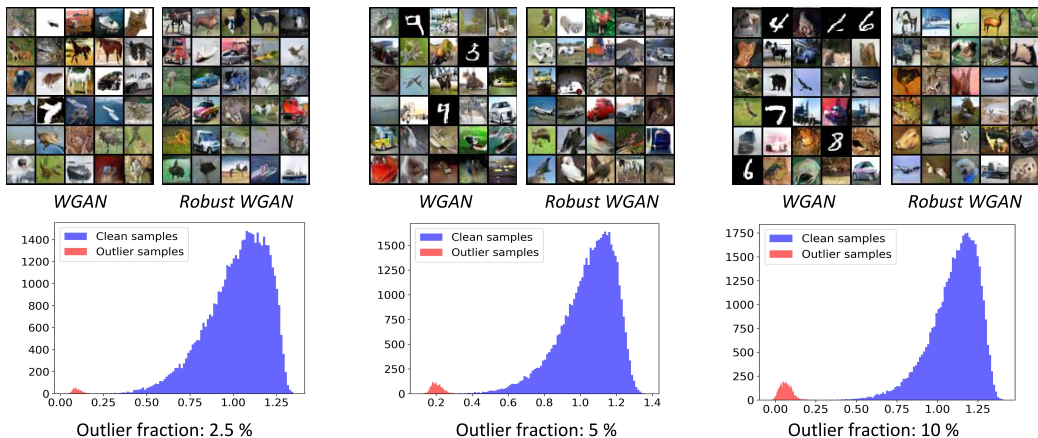

Figure 3: Visualizing samples and weight histograms. In the top panel, we show samples generated by WGAN and robust WGAN trained on the CIFAR-10 dataset corrupted with MNIST samples as outliers. WGAN fits both CIFAR and MNIST samples, while the robust WGAN ignores the outliers. In the bottom panel, we visualize the weights (output of the $\mathbf{W}(.)$ function) for in-distribution and outlier samples. The outlier samples are assinged low weights while in-distribution samples get large weights.

## 4.1 Generative modeling

In this section, we show how our robust Wasserstein formulation can be used to train GANs that are insensitive to outliers. The core idea is to train a GAN by minimizing the robust Wasserstein measure (in dual form) between real and generative data distributions. Let $\mathbf{G}$ denote a generative model which maps samples from random noise vectors to real data distribution. Using the one-directional version of the dual form of robust Wasserstein measure (8), we obtain the following optimization problem

$$\min_{\mathbf{W},\mathbf{G}} \; \max_{\mathbf{D}\in Lip-1} \mathbb{E}_{\mathbf{x}\sim\mathbb{P}_X}[\mathbf{W}(\mathbf{x})\mathbf{D}(\mathbf{x})] - \mathbb{E}_{\mathbf{z}}[\mathbf{D}(\mathbf{G}(\mathbf{z}))]$$

$$\text{s.t} \quad \mathbb{E}_{\mathbf{x}\sim\mathbb{P}_X}[(\mathbf{W}(\mathbf{x})-1)^2] \leq 2\rho, \;\; \mathbb{E}_{\mathbf{x}\sim\mathbb{P}_X}[\mathbf{W}(\mathbf{x})] = 1, \;\; \mathbf{W}(\mathbf{x}) \geq 0$$

The first constraint is imposed using a Lagrangian term in the objective function. To impose the second constraint, we use ReLU as the final layer of $\mathbf{W}(.)$ network and normalize the weights by the sum of weight vectors in a batch. This leads to the following optimization

$$\min_{\mathbf{W},\mathbf{G}} \max_{\mathbf{D}\in Lip-1} \mathbb{E}_{\mathbf{x}}[\mathbf{W}(\mathbf{x})\mathbf{D}(\mathbf{x})] - \mathbb{E}_{\mathbf{z}}[\mathbf{D}(\mathbf{G}(\mathbf{z}))] + \lambda \max\left(\mathbb{E}_{\mathbf{x}}[(\mathbf{W}(\mathbf{x})-1)^2] - 2\rho, 0\right) \quad (9)$$

We set $\lambda$ to a large value (typically $\lambda = 1000$) to enforce the constraint on $\chi^2$-divergence. A detailed algorithm can be found in Appendix. Our robust Wasserstein formulation can easily be extended to other GAN objective functions such as non-saturating loss and hinge loss, as discussed in Appendix.

**Datasets with outliers:** First, we train the robust Wasserstein GAN on datasets corrputed with outlier samples. For the ease of quantitative evaluation, the outlier corrupted dataset is constructed as

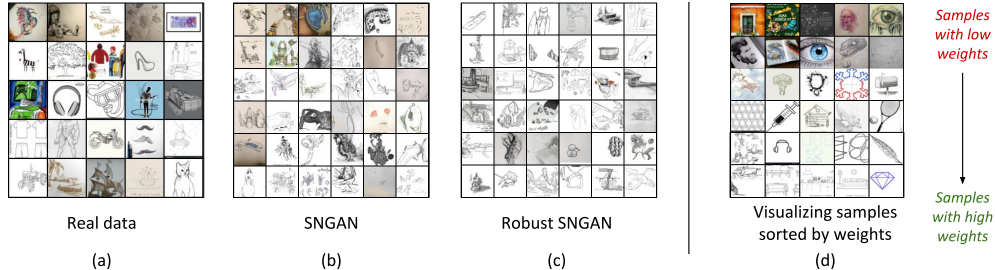

Figure 4: Visualizing samples generated on Domainnet sketch dataset. In panels (a), (b) and (c), we show the real data, samples generated by SNGAN and robust SNGAN, respectively. Robust SNGAN only generates images of sketches ignoring outliers. In panel (d), we visualize real samples sorted by weights. Low weights are assigned to outliers, while sketch images get large weights.

Table 1: Quantitative evaluation of robust WGAN on clean datasets. In each cell, the top row corresponds to the Inception score and the bottom row corresponds to the FID score.

| Dataset | Arch | WGAN | RWGAN $\rho = 0$ | RWGAN $\rho = 0.3$ |
|---|---|---|---|---|
| CIFAR-10 | DCGAN | 6.86 28.46 | 6.84 29.11 | 6.91 29.45 |
| CIFAR-10 | Resnet | 7.49 21.73 | 7.35 21.98 | 7.36 21.57 |
| CIFAR-100 | Resnet | 9.01 15.60 | 8.79 15.61 | 8.93 15.32 |

Table 2: Cross-domain recognition accuracy on VISDA-17 dataset using Resnet-18 model averaged over 3 runs.

| Method | Accuracy (in %) |
|---|---|
| Source only | 44.7 |
| Adversarial (*no ent*) | 55.4 |
| Robust adversarial (*no ent*) | 62.9 |
| Adversarial (*with ent*) | 59.5 |
| Robust adversarial (*with ent*) | **63.9** |

follows: We artificially add outlier samples to the CIFAR-10 dataset such they occupy $\gamma$ fraction of the samples. MNIST and uniform noise are used as two choices of outlier distributions. Samples generated by Wasserstein GAN and robust Wasserstein GAN on this dataset are shown in Fig. 3. While Wasserstein GAN fits outliers in addition to the CIFAR samples, the robust Wasserstein GAN effectively ignores outliers and generates samples only from the CIFAR-10 dataset.

For a quantitative evaluation, we report the FID scores of the generated samples with respect to the clean CIFAR-10 distribution (Figure 2). Since Wasserstein GAN generates outlier samples in addition to the CIFAR-10 samples, the FID scores get worse as the outlier fraction increases. Robust Wasserstein GAN, on the other hand, obtains good FID even for large fraction of outliers. This trend is consistent for both outlier distributions MNIST and uniform noise.

Next, we train our robust GAN model on a dataset where outliers are naturally present. We use *Sketch* domain of DomainNet dataset [19] for this purpose. As shown in Figure 4(a), the dataset contains many outlier samples (non-sketch images). Samples generated by spectral normalization GAN and robust spectral normalization GAN (both using Resnet) are shown in Figure 4(b, c). We observe that the SNGAN model generates some non-sketch images in addition to sketch images. Robust SNGAN, on the other hand, ignores outliers and only generates samples that look like sketches.

**Clean datasets:** In the previous section, we demonstrated how robust Wasserstein GAN effectively ignores outliers in the data distributions. A natural question that may arise is what would happen if one uses the robust WGAN on a clean dataset (dataset without outliers). To understand this, we train the robust Wasserstein GAN on CIFAR-10 and CIFAR-100 datasets. The Inception and FID scores of generated samples are reported in Table. 1. We observe no drop in FID scores, which suggest that no modes are dropped in the generated distribution.

**Usefulness of sample weights:** In the optimization of the robust GAN, each sample is assigned a weight indicating the difficulty of that sample to be generated by the model. In this section, we visualize the weights learnt by our robust GAN. In Figure 3, we plot the histogram of weights assigned to in-distribution and outlier samples for robust WGAN trained on CIFAR-10 dataset corrupted with MNIST outliers. Outliers are assigned smaller weights compared to the in-distribution samples, and

Table 3: Adaptation accuracy on VISDA-17 using Resnet-50 model averaged over 3 runs

| | Method | Accuracy (in %) |
|---|---|---|
| | Source Only | 50.7 |
| | DAN [14] | 53.0 |
| | RTN [16] | 53.6 |
| | DANN [10] | 55.0 |
| | JAN-A [17] | 61.6 |
| | GTA [23] | 69.5 |
| | SimNet [21] | 69.6 |
| | CDAN-E [15] | 70.0 |
| *Ours* | Adversarial (*no ent*) | 62.9 |
| | Robust adversarial (*no ent*) | 68.6 |
| | Adversarial (*with ent*) | 65.5 |
| | Robust adversarial (*with ent*) | **71.5** |

Table 4: Adaptation accuracy on VISDA-17 using Resnet-101 model averaged over 3 runs

| | Method | Accuracy (in %) |
|---|---|---|
| | Source only | 55.3 |
| | DAN [14] | 61.1 |
| | DANN [10] | 57.4 |
| | MCD [22] | 71.9 |
| *Ours* | Adversarial (*no ent*) | 65.5 |
| | Robust adversarial (*no ent*) | 69.3 |
| | Adversarial (*with ent*) | 69.3 |
| | Robust adversarial (*with ent*) | **72.7** |

Table 5: Sensitivity Analysis of $\rho$

| **GAN exp** | $\rho$ | 0 | 0.01 | 0.05 | 0.1 | 0.15 |
|---|---|---|---|---|---|---|
| CIFAR + MNIST | FID | 37.5 | 34.7 | 31.9 | **29.9** | 30.2 |

| **DA exp** | $\rho$ | 0.0 | 0.05 | 0.1 | 0.2 | 0.4 |
|---|---|---|---|---|---|---|
| Resnet-18 | Acc | 59.5 | 62.8 | 63.1 | **63.9** | 63.6 |

there is a clear separation between their corresponding histograms. For the GAN model trained on the Sketch dataset, we show a visualization of randomly chosen input samples sorted by their assigned weights in Figure 4(d). We observe that non-sketch images are assigned low weights while the true sketch images obtain larger weights. Hence, the weights learnt by our robust GAN can be a useful indicator for assessing how difficult it is to generate a given sample.

## 4.2 Domain adaptation

In Unsupervised Domain Adaptation (UDA) problem, we are given a labeled source dataset and an unlabeled target dataset. The source and target domains have a covariate shift i.e., the conditional distribution of the data given labels differ while the marginal distribution of labels match. Due to the covariate shift, a model trained solely on the source domain performs poorly on the target. A conventional approach for UDA involves training classification model on the source domain while minimizing a distributional distance between source and target feature distributions. Commonly used distance measures include Wasserstein distance [24] and non-saturating loss [10]. For the ease of explanation, we use Wasserstein as our choice of distance measure.

Let $\mathbb{P}_s = \{(\mathbf{x}_i^s, y_i^s)\}_{i=1}^{n_s}$ and $\mathbb{P}_t = \{(\mathbf{x}_i^t)\}_{i=1}^{n_t}$ denote the source and target distributions, respectively. Let $\mathbf{F}$ denote a feature network, and $\mathbf{C}$ denote a classifier. Then, the UDA optimization that minimizes the robust OT distance between source and target feature distributions can be written as

$$\min_{\mathbf{F},\mathbf{C}} \frac{1}{n_s} \sum_i \mathcal{L}_{cls}(\mathbf{F}(\mathbf{x}_i), y_i) + \lambda \left[ \min_{\mathbf{w} \in \Delta^{n_t}} \max_{\mathbf{D}} \frac{1}{n_s} \sum_i \mathbf{D}(\mathbf{F}(\mathbf{x}_i^s)) - \frac{1}{n_t} \sum_j w_j \mathbf{D}(\mathbf{F}(\mathbf{x}_j^t)) \right] \quad (10)$$

$$\text{s.t } \|n_t\mathbf{w} - 1\|_2 \leq \sqrt{2\rho n_t}$$

where $\mathbf{w} = [w_1, w_2 \ldots w_{n_t}]$. While we describe this formulation for the Wasserstein distance, similar ideas can be applied to other adversarial losses. For instance, by replacing the second and third terms of (10) with binary cross entropy loss, we obtain the non-saturating objective. Note that we use the discrete formulation of dual objective (Section 3.2) instead of the continuous one (Section 3.2). This is because in our experiments, small batch sizes ($\sim 28$) were used due to GPU limitations. With small batch sizes, continuous relaxation gives sub-optimal performance.

For experiments, we use VISDA-17 dataset [20], which is a large scale benchmark dataset for UDA. The task is to perform 12- class classification by adapting models from synthetic to real dataset. In our experiments, we use non-saturating loss instead of Wasserstein to enable fair comparison with other adversarial approaches such as DANN. In addition to the adversarial alignment, we use an entropy regularizer on target logits, which is a standard technique used in UDA [5]. The adaptation results using Resnet-18, Resnet-50 and Resnet-101 models are shown in Tables 2, 3 and 4, respectively. Our robust adversarial objective gives consistent performance improvement of $\sim 5\%$ over the standard adversarial objective in all experiments. By using a weighted adversarial loss, our approach assigns

low weights to samples that are hard to adapt and high weights to target samples that look more similar to source, thereby promoting improved adaptation. Also, with the use of entropy regularization, our generic robust adversarial objective reaches performance on par with other competing approaches that are tuned specifically for the UDA problem. This demonstrates the effectiveness of our approach.

**Ablation: Sensitivity of** $\rho$   In Table. 5, we report the sensitivity of $\rho$ for both GANs and domain adaptation experiments. In the case of GANs, performance is relatively low only for very low values of $\rho$ and stable for higher values. For DA, sensititivity is low in general. For all DA experiments, we used $\rho = 0.2$ without tuning it individually for each setting.

## 5   Conclusion

In this work, we study the robust optimal transport which is insensitive to outliers (samples with large noise) in the data. The applications of previous formulations of robust OT are limited in practical deep learning problems such as GANs and domain adaptation due to the instability of their optimization solvers. In this paper, we derive a computationally efficient dual form of the robust OT objective that is suited for deep learning applications. We demonstrate the effectiveness of the proposed method in two applications of GANs and domain adaptation, where our approach is shown to effectively handle outliers and achieve good performance improvements.

## 6   Broader Impact

The use of optimal transport (OT) distances such as the Wasserstein distance have become increasingly popular in machine learning with several applications in generative modeling, image-to-image translation, inpainting, domain adaptation, etc. One of the shortcomings of OT is its sensitivity to input noise. Hence, using OT for large-scale machine learning problems can be problematic since noise in large datasets is inevitable. Building on theoretical formulations of unbalanced OT which suffer from computational instability in deep learning applications, we have developed an efficient learning method that is provably robust against outliers and is amenable to complex deep learning applications such as deep generative modeling and domain adaptation. These attributes ensure broader impacts of this work in both theoretical and applied machine learning communities and can act as a bridge between the two. To the best of our knowledge, this work does not lead to any negative outcomes either in ethical or societal aspects.

## 7   Acknowledgements

This project was supported in part by NSF CAREER AWARD 1942230, DOE Award 302629-00001, grants from Capital One and Qualcomm, a Simons Fellowship on Deep Learning Foundations, and a MURI program from the Army Research Office under the grant W911NF17-1-0304.

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
