[Supplementary Material]

# Supplementary Material: Robust Optimal Transport with Applications in Generative Modeling and Domain Adaptation

## 1 Proofs

In this section, we present the proofs:

**Duality:**

**Theorem 1.** *Let $\mathbb{P}_X$ and $\mathbb{P}_Y$ be two distributions defined on a metric space. The robust Wasserstein measure admits the following dual form*

$$\mathcal{W}^{rob}_{\rho_1,\rho_2}(\mathbb{P}_X, \mathbb{P}_Y) = \min_{\mathbb{P}_{\tilde{X}}, \mathbb{P}_{\tilde{Y}}} \max_{k \in Lip-1} \int D(x) d\mathbb{P}_{\tilde{X}} - \int D(x) d\mathbb{P}_{\tilde{Y}} \tag{1}$$

$$s.t. \ \mathcal{D}_f(\mathbb{P}_{\tilde{X}}||\mathbb{P}_X) \leq \rho_1, \ \mathcal{D}_f(\mathbb{P}_{\tilde{Y}}||\mathbb{P}_Y) \leq \rho_2.$$

**Proof:** We begin with the primal form of the robust optimal transport defined as

$$\mathcal{W}^{rob}_{\rho_1,\rho_2}(\mathbb{P}_X, \mathbb{P}_Y) = \min_{\mathbb{P}_{\tilde{X}}, \mathbb{P}_{\tilde{Y}} \in Prob(\mathcal{X})} \min_{\pi} \int \int c(x,y)\pi(x,y)dxdy$$

$$s.t. \ \mathcal{D}_f(\mathbb{P}_{\tilde{X}}||\mathbb{P}_X) \leq \rho_1, \ \mathcal{D}_f(\mathbb{P}_{\tilde{Y}}||\mathbb{P}_Y) \leq \rho_2$$

$$\int \pi(x.y)dy = \mathbb{P}_{\tilde{X}}, \int \pi(x.y)dx = \mathbb{P}_{\tilde{Y}}$$

The constraint $\mathbb{P}_{\tilde{X}}, \mathbb{P}_{\tilde{Y}} \in Prob(\mathcal{X})$ states that $\mathbb{P}_{\tilde{X}}$ and $\mathbb{P}_{\tilde{Y}}$ are valid probability distributions. For brevity, we shall ignore explicitly stating it in the rest of the proof. Now, we write the Lagrangian function with respect to marginal constraints.

$$L = \min_{\mathbb{P}_{\tilde{X}}, \mathbb{P}_{\tilde{Y}}} \min_{\pi > 0} \max_{\phi(x), \psi(y)} \int \int c(x,y)\pi(x,y)dxdy + \int \phi(x) \left( \int \mathbb{P}_{\tilde{X}} - \pi(x,y)dy \right) dx$$

$$+ \int \psi(y) \left( \int \pi(x,y)dx - \mathbb{P}_{\tilde{Y}} \right) dy$$

$$s.t. \ \mathcal{D}_f(\mathbb{P}_{\tilde{X}}||\mathbb{P}_X) \leq \rho_1, \ \mathcal{D}_f(\mathbb{P}_{\tilde{Y}}||\mathbb{P}_Y) \leq \rho_2$$

$$= \min_{\mathbb{P}_{\tilde{X}}, \mathbb{P}_{\tilde{Y}}} \min_{\pi > 0} \max_{\phi(x), \psi(y)} \int \int [c(x,y) - \phi(x) + \psi(y)] \pi(x,y)dxdy + \int \phi(x)d\mathbb{P}_{\tilde{X}} - \int \psi(y)d\mathbb{P}_{\tilde{Y}}$$

$$s.t. \ \mathcal{D}_f(\mathbb{P}_{\tilde{X}}||\mathbb{P}_X) \leq \rho_1, \ \mathcal{D}_f(\mathbb{P}_{\tilde{Y}}||\mathbb{P}_Y) \leq \rho_2$$

Since $\pi > 0$, we observe that

$$c(x,y) - \phi(x) + \psi(y) = \begin{cases} \infty & \text{if } c(x,y) - \phi(x) + \psi(y) > 0 \\ 0 & \text{otherwise} \end{cases}$$

Hence, the dual formulation becomes

$$\mathcal{W}^{rob}_{\rho_1,\rho_2}(\mathbb{P}_X,\mathbb{P}_Y) = \min_{\mathbb{P}_{\tilde{X}},\mathbb{P}_{\tilde{Y}}} \max_{\phi(x),\psi(y)} \int \phi(x)d\mathbb{P}_{\tilde{X}} - \int \psi(y)d\mathbb{P}_{\tilde{Y}} \tag{2}$$
$$\text{s.t. } \phi(x) - \psi(y) \le c(x,y)$$
$$\mathcal{D}_f(\mathbb{P}_{\tilde{X}}||\mathbb{P}_X) \le \rho_1,\ \mathcal{D}_f(\mathbb{P}_{\tilde{Y}}||\mathbb{P}_Y) \le \rho_2$$

Furthermore, when the distributions lie in a metric space, we can further simplify this duality. Define

$$k(x) := \inf_y c(x,y) + \psi(y) \tag{3}$$

Since the feasible set in the dual problem satisfies $\phi(x) - \psi(y) \le c(x,y)$, $\phi(x) \le k(x)$, and by using $y = x$ in Eq (3), we obtain, $k(x) \le \psi(x)$. Hence, $\phi(x) \le k(x) \le \psi(x)$.

$$|k(x) - k(x')| = |\inf_y[c(x,y) + \psi(y)] - \inf_y[c(x',y) + \psi(y)]|$$
$$\le |c(x,y) - c(x',y)|$$

Hence, $k(.)$ is 1-Lipschitz. Using the above inequalities in (2), we obtain,

$$\mathcal{W}^{rob}_{\rho_1,\rho_2}(\mathbb{P}_X,\mathbb{P}_Y) \le \min_{\mathbb{P}_{\tilde{X}},\mathbb{P}_{\tilde{Y}}} \max_{k\in Lip-1} \int k(x)d\mathbb{P}_{\tilde{X}} - \int k(x)d\mathbb{P}_{\tilde{Y}}$$
$$\text{s.t. } \mathcal{D}_f(\mathbb{P}_{\tilde{X}}||\mathbb{P}_X) \le \rho_1,\ \mathcal{D}_f(\mathbb{P}_{\tilde{Y}}||\mathbb{P}_Y) \le \rho_2$$

Also, $\phi(x) = k(x)$ and $\psi(x) = k(x)$ is a feasible solution in optimization (2). Since (2) maximizes over $\phi(.)$ and $\psi(.)$, we obtain

$$\mathcal{W}^{rob}_{\rho_1,\rho_2}(\mathbb{P}_X,\mathbb{P}_Y) \ge \min_{\mathbb{P}_{\tilde{X}},\mathbb{P}_{\tilde{Y}}} \max_{k\in Lip-1} \int k(x)d\mathbb{P}_{\tilde{X}} - \int k(x)d\mathbb{P}_{\tilde{Y}}$$
$$\text{s.t } \mathcal{D}_f(\mathbb{P}_X,\mathbb{P}_{\tilde{X}}) \le \rho_1,\ \mathcal{D}_f(\mathbb{P}_Y,\mathbb{P}_{\tilde{Y}}) \le \rho_2$$

Combining these two inequalities, we obtain

$$\mathcal{W}^{rob}_{\rho_1,\rho_2}(\mathbb{P}_X,\mathbb{P}_Y) = \min_{\mathbb{P}_{\tilde{X}},\mathbb{P}_{\tilde{Y}}} \max_{k\in Lip-1} \int k(x)d\mathbb{P}_{\tilde{X}} - \int k(x)d\mathbb{P}_{\tilde{Y}} \tag{4}$$
$$\text{s.t } \mathcal{D}_f(\mathbb{P}_X,\mathbb{P}_{\tilde{X}}) \le \rho_1,\ \mathcal{D}_f(\mathbb{P}_Y,\mathbb{P}_{\tilde{Y}}) \le \rho_2$$

The above equation is similar in spirit to the *Kantrovich-Rubinstein* duality. An important observation to note is that the above optimization only maximizes over a single discriminator function (as opposed to two functions in optimization (2)). Hence, it is easier to train it in large-scale deep learning problems such as GANs.

**Provable robustness:**

**Theorem 2.** *Let $\mathbb{P}_X$ and $\mathbb{P}_Y$ be two distributions such that $\mathbb{P}_X$ is corrupted with $\gamma$ fraction of outliers i.e., $\mathbb{P}_X = (1-\gamma)\mathbb{P}^c_X + \gamma\mathbb{P}^a_X$, where $\mathbb{P}^c_X$ is the clean distribution and $\mathbb{P}^a_X$ is the outlier distribution. Let $\mathcal{W}(\mathbb{P}^a_X,\mathbb{P}^c_X) = k\mathcal{W}(\mathbb{P}^c_X,\mathbb{P}_Y)$, with $k \ge 1$. Then,*

$$\mathcal{W}^{rob}_{\rho,0}(\mathbb{P}_X,\mathbb{P}_Y) \le \max\left(1, 1 + k\gamma - k\sqrt{2\rho\gamma(1-\gamma)}\right)\mathcal{W}(\mathbb{P}^c_X,\mathbb{P}_Y).$$

**Proof:** We consider the case of empirical distributions. Let $\{\mathbf{x}^a_i\}^{n_a}_{i=1}$ be the samples in the anomaly distribution $\mathbb{P}^a_X$, $\{\mathbf{x}^c_i\}^{n_c}_{i=1}$ be the samples in the clean distribution $\mathbb{P}^c_X$, and $\{\mathbf{y}_i\}^m_{i=1}$ be the samples in the distribution $\mathbb{P}_Y$. We also know that $\frac{n_a}{n_a+n_c} = \gamma$.

$\mathcal{W}^{rob}_{\rho,0}(\mathbb{P}_X,\mathbb{P}_Y)$ is defined as

$$\mathcal{W}^{rob}_{\rho,0}(\mathbb{P}_X,\mathbb{P}_Y) = \min_{\mathbb{P}_{\tilde{X}}\in Prob(\mathcal{X})} \min_{\pi} \sum_i \sum_j \pi_{ij}c_{ij}$$
$$\text{s.t. } \sum_j \pi_{ij} = \mathbb{P}_{\tilde{X}},\quad \sum_i \pi_{ij} = \mathbb{P}_Y,\ \mathcal{D}_{\chi^2}(\mathbb{P}_{\tilde{X}}||\mathbb{P}_X) \le \rho$$

Let $\pi^{c*}$ and $\pi^{a*}$ be the optimal transport plans for $\mathcal{W}(\mathbb{P}_X^c, \mathbb{P}_Y)$ and $\mathcal{W}(\mathbb{P}_X^a, \mathbb{P}_Y)$ respectively. We consider transport plans of the form $\beta\pi^{a*} + (1-\beta)\pi^{c*}$, for $\beta \in [0, 1]$. The marginal constraints can then be written as

$$\int \beta\pi^{a*} + (1-\beta)\pi^{c*} dx = \beta\mathbb{P}_X^a + (1-\beta)\mathbb{P}_X^c$$

$$\int \beta\pi^{a*} + (1-\beta)\pi^{c*} dy = \mathbb{P}_Y$$

For this to be a feasible solution for $\mathcal{W}_{\rho,0}^{rob}(\mathbb{P}_X, \mathbb{P}_Y)$, we require

$$\mathcal{D}_{\chi^2}(\beta\mathbb{P}_X^a + (1-\beta)\mathbb{P}_X^c || \gamma\mathbb{P}_X^a + (1-\gamma)\mathbb{P}_X^c) \leq \rho$$

The distribution $\beta\mathbb{P}_X^a + (1-\beta)\mathbb{P}_X^c$ can be characterized as

$$[\underbrace{\frac{\beta}{n_a}, \cdots}_{n_a \text{ terms}}, \underbrace{\frac{1-\beta}{n_c}, \cdots}_{n_c \text{ terms}}].$$

Using this, the above constraint can be written as

$$(\beta - \gamma)^2 \leq 2\rho\gamma(1-\gamma) \tag{5}$$

Hence, all transport plans of the form $\beta\pi^{a*} + (1-\beta)\pi^{c*}$ are feasible solutions of $\mathcal{W}_{\rho,0}^{rob}(\mathbb{P}_X, \mathbb{P}_Y)$ if $\beta$ satisfies $(\beta - \gamma)^2 \leq 2\rho\gamma(1-\gamma)$. Therefore, we have:

$$\mathcal{W}_{\rho,0}^{rob}(\mathbb{P}_X, \mathbb{P}_Y) \leq \min_\beta \sum_i \sum_j c_{i,j}[\beta\pi_{i,j}^{a*} + (1-\beta)\pi_{i,j}^{c*}]$$

$$\leq \min_\beta \beta\mathcal{W}(\mathbb{P}_X^a, \mathbb{P}_Y) + (1-\beta)\mathcal{W}(\mathbb{P}_X^c, \mathbb{P}_Y)$$

$$\text{s.t. } (\beta - \gamma)^2 \leq 2\rho\gamma(1-\gamma)$$

By the assumption, we have:

$$\mathcal{W}(\mathbb{P}_X^c, \mathbb{P}_X^a) = k\mathcal{W}(\mathbb{P}_X^c, \mathbb{P}_Y)$$
$$\mathcal{W}(\mathbb{P}_X^a, \mathbb{P}_Y) \leq \mathcal{W}(\mathbb{P}_X^a, \mathbb{P}_X^c) + \mathcal{W}(\mathbb{P}_X^c, \mathbb{P}_Y)$$
$$\leq (k+1)\mathcal{W}(\mathbb{P}_X^c, \mathbb{P}_Y)$$

Hence,

$$\mathcal{W}_{\rho,0}^{rob}(\mathbb{P}_X, \mathbb{P}_Y) \leq \min_\beta (1 + \beta k)\mathcal{W}(\mathbb{P}_X^c, \mathbb{P}_Y)$$

$$\text{s.t. } (\beta - \gamma)^2 \leq 2\rho\gamma(1-\gamma)$$

The smallest value $\beta$ can take is $\gamma - \sqrt{2\rho\gamma(1-\gamma)}$. This gives

$$\mathcal{W}_{\rho,0}^{rob}(\mathbb{P}_X, \mathbb{P}_Y) \leq \max\left(1, 1 + k\gamma - k\sqrt{2\rho\gamma(1-\gamma)}\right)\mathcal{W}(\mathbb{P}_X^c, \mathbb{P}_Y)$$

**Note:** The transport plan $\beta\pi^{a*} + (1-\beta)\pi^{c*}$ is not the the optimal transport plan for the robust OT optimization between corrupted distribution $\mathbb{P}_X$ and $\mathbb{P}_Y$. However, this plan is a "feasible" solution that satisfies the constraints of the robust OT. Hence, the cost obtained by this plan is an upper bound to the true robust OT cost.

## 2 Practical Issues with Unbalanced Optimal Transport Dual

### 2.1 Dual Objective

The primal form of unbalanced OT [2, 1] is given by

$$\mathcal{W}^{ub}(\mathbb{P}_X, \mathbb{P}_Y) = \min_{\pi \in \Pi(\mathbb{P}_{\tilde{X}}, \mathbb{P}_{\tilde{Y}})} \int c(x,y)\pi(x,y)dxdy + \mathcal{D}_f(\mathbb{P}_{\tilde{X}} || \mathbb{P}_X) + \mathcal{D}_f(\mathbb{P}_{\tilde{Y}} || \mathbb{P}_Y) \tag{6}$$

where $\mathcal{D}_f$ is the $f$-divergence between distributions, defined as $\mathcal{D}_f(P||Q) = \int f(\frac{dP}{dQ})dQ$. Furthermore, the authors of [2] derived a dual form for the problem. Let $f$ be a convex lower semi-continuous function. Define $r^*(x) := \sup_{s>0} \frac{x - f(s)}{s}$ where $f'_\infty := \lim_{s\to\infty} \frac{f(s)}{s}$. Then,

$$\mathcal{W}^{ub}(\mathbb{P}_X, \mathbb{P}_Y) = \max_{\phi,\psi} \int \phi(x)d\mathbb{P}_X + \int \psi(y)d\mathbb{P}_Y \qquad (7)$$
$$\text{s.t. } r^*(\phi(x)) + r^*(\psi(y)) \le c(x,y)$$

In our case, $f(x) = (x-1)^2/2$. Let us now, write the dual for this case. First, we need to find $r^*(x)$ for this function.

$$r^*(x) = \sup_{s>0} \frac{x - f(s)}{s}$$
$$= \sup_{s>0} \frac{x - (s-1)^2/2}{s}$$
$$= \sup_{s>0} \frac{2x - 1 - s^2 + 2s}{2s}$$
$$= \sup_{s>0} \frac{2x - 1}{2s} - \frac{s}{2} + 1$$

We consider three cases:

**Case 1:** $x > 1/2$.　　In this case, $r^*(x) \to \infty$ as $s \to 0^+$.

**Case 2:** $x = 1/2$.　　In this case, $r^*(x) = 1$.

**Case 3:** $x < 1/2$.　　In this case, for $s \to 0^+$, $r^*(x) \to -\infty$. Also, when $s \to \infty$, $r^*(x) \to -\infty$. So, the maximizer has to lie somewhere in $(0, \infty)$.

Taking the derivative w.r.t. $s$, we obtain,

$$\frac{1 - 2x}{2s^2} - \frac{1}{2} = 0$$

This gives

$$s = \sqrt{1 - 2x}$$

The second derivative is also negative at this point. Hence, it is the maximizer point. Substituting this in $r^*(\cdot)$, we obtain

$$r^*(x) = 1 - \sqrt{1 - 2x}$$

This gives the solution,

$$r^*(x) = \begin{cases} \infty & x > 1/2 \\ 1 - \sqrt{1 - 2x} & x \le 1/2 \end{cases}$$

Hence, the dual form becomes

$$\mathcal{W}^{ub}(\mathbb{P}_X, \mathbb{P}_Y) = \max_{\phi,\psi} \int \phi(x)d\mathbb{P}_X + \int \psi(y)d\mathbb{P}_Y \qquad (8)$$
$$\text{s.t. } 1 - \sqrt{1 - 2\phi(x)} + 1 - \sqrt{1 - 2\psi(y)} \le c(x,y)$$
$$\phi(x) \le 1/2, \ \psi(y) \le 1/2$$

(a) Our dual             (b) Unbalanced Optimal Transport dual

Figure 1: Loss curves of training GANs using practical implementations of our proposed dual vs. the dual of Unbalanced OT. Our model converges to a proper solution while the model trained using the dual of unbalanced OT does not converge to a good solution.

## 2.2 Practical Implementation and Training Issues

To simplify notation, we make a substitution $\mathbf{D}_1(x) \leftarrow 2\phi(x)$ and $\mathbf{D}_2(x) \leftarrow 2\psi(x)$ in (8). This gives

$$\mathcal{W}^{ub}(\mathbb{P}_X, \mathbb{P}_Y) = \max_{\mathbf{D}_1, \mathbf{D}_2} \int \mathbf{D}_1(x) d\mathbb{P}_X + \int \mathbf{D}_2(y) d\mathbb{P}_Y \tag{9}$$
$$\text{s.t. } 1 - \sqrt{1 - \mathbf{D}_1(x)} + 1 - \sqrt{1 - \mathbf{D}_2(y)} \leq c(x,y)$$
$$\mathbf{D}_1(x) \leq 1, \ \mathbf{D}_2(y) \leq 1$$

Then, the second set of constraints (i.e. $\mathbf{D}_1(x)$ and $\mathbf{D}_2(y)$ to be less or equal to one) can be integrated into the network design. The first constraint can be implemented using a Lagrangian constraint. This leads us to the following objective:

$$\min_{\mathbf{G}} \max_{\mathbf{D}_1, \mathbf{D}_2 \leq 1} \mathbb{E}_{\mathbf{x} \sim p_{data}} \mathbf{D}_1(\mathbf{x}) + \mathbb{E}_{\mathbf{z} \sim p_z} \mathbf{D}_2(\mathbf{G}(\mathbf{z})) \tag{10}$$
$$- \lambda \mathbb{E}_{\mathbf{x}, \mathbf{z}} \left[ \max \left( \sqrt{1 - \mathbf{D}_1(\mathbf{x})} + \sqrt{1 - \mathbf{D}_2(\mathbf{G}(\mathbf{z}))} + c(\mathbf{x}, \mathbf{G}(\mathbf{z})) - 2, 0 \right) \right]$$

The constraint $\mathbf{D}_1, \mathbf{D}_2 \leq 1$ means $\mathbf{D}_1(\mathbf{x}) \leq 1, \mathbf{D}_2(\mathbf{x}) \leq 1, \forall \mathbf{x}$. We trained a model using this dual objective on CIFAR-10 dataset using Resnet architecture for generator and discrminator network. However, partially due to the presence of two discriminator networks $\mathbf{D}_1$ and $\mathbf{D}_2$, the training is challenging. Similar training difficulties in GANs with multiple discriminator networks have been observed in [4]. Even with a sweep of hyper-parameters, we were not able to make the model based on optimization 10 to converge to a proper solution.

Training loss curves of our model vs. that of unbalanced OT are shown in Figure. 1. Samples generated by our dual and unbalanced OT dual is shown in Figure. 2. We observe that the model trained using the Unbalanced OT dual produces loss curve that is flat and does not learn a proper solution. This is also evident from Figure. 2. Models trained using our dual generates CIFAR-like samples, while the one trained with unbalanced OT dual produces noisy images.

## 3 Choosing $\rho$ and the tightness of the bound

The constant $\rho$ in our formulation is a hyper-parameter that needs to be estimated. The value of $\rho$ denotes the extent of marginal relaxation. In applications such as GANs or domain adaptation, performance on a validation set can be used for choosing $\rho$. In the absence of validation set, we present two techniques for estimating $\rho$.

**When outlier fraction is known:** When the outlier fraction $\gamma$ is known, a good estimate of $\rho$ is $\rho = \gamma/2(1 - \gamma)$. Below we explain this claim.

| (a) Our dual | (b) Unbalanced Optimal Transport dual |

Figure 2: Samples generated by our dual and Unbalanced OT dual.

Ideally, we desire the perturbed distribution $\mathbb{P}_{\tilde{X}}$ to be such that

$$\mathbb{P}_{\tilde{X}} = [\underbrace{\kappa, \kappa, \ldots \kappa}_{\text{normal samples}}, \underbrace{0, 0, \ldots 0}_{\text{outlier samples}}].$$

Since the outlier fraction is $\gamma$, the number of normal samples is $(1-\gamma)n$ and the number of outlier samples is $\gamma n$. For $\mathbb{P}_{\tilde{X}}$ to be a valid pmf, we require,

$$\kappa(1-\gamma)n = 1$$

$$\kappa = \frac{1}{(1-\gamma)n}$$

Also, we have a constraint on $f$-divergence to ensure $\mathcal{D}_{\chi^2}(\mathbb{P}_{\tilde{X}}||\mathbb{P}_X) \leq \rho$. This condition simplifies to

$$\frac{1}{2}\left[\frac{(\alpha - 1/n)^2}{1/n}(1-\gamma)n + \frac{(0 - 1/n)^2}{1/n}\gamma n\right] \leq \rho$$

Therefore, we obtain

$$\rho \geq \frac{\gamma}{1-\gamma}$$

Hence, to get a proper estimate of the robust Wasserstein distance, we choose $\rho = \gamma/2(1-\gamma)$. Note that by substituting $\rho = \gamma/2(1-\gamma)$ in Theorem 1, we obtain, $\mathcal{W}_{\rho,0}^{rob}(\mathbb{P}_X, \mathbb{P}_Y) \leq \mathcal{W}(\mathbb{P}_X, \mathbb{P}_Y)$.

**Heuristic:** In general, estimating $\rho$ is non-trivial. We now present a heuristic that can be used for this purpose. First, we compute robust OT measure for various values of $\rho$. This curve has an elbow shape, and its point of inflection can be used as an estimate of $\rho$. We demonstrate this with an example. We use the mixture of four Gaussians datasets as shown in Figure 1 of the main paper as our input distributions. The means of the Gaussians are placed in a circumference of a circle, with two distributions being the rotated versions of each other. We introduce 5% outlier samples in one distribution. A plot of robust Wasserstein measure varying $\rho$ is shown in Fig. 3 (in blue). We observe that initially as $\rho$ increases, the robust OT value decreases sharply followed by a gradual descent, resembling the pattern of the elbow curve in $k$-means clustering. The point of inflection in this elbow curve is a proper estimate of $\rho$.

**Tightness of the upper bound:** In Figure. 3, we plot the upper bound of the robust OT estimate as given by Theorem 2 in red. We observe that the upper bound is fairly tight in this case. It closely approximates the true robust OT measure given by the blue curve.

## 4 Biases And Mode Drop

The model of outliers we assume in our paper is that of large noise (Section 3). Hence, our model drops samples that are far from the true data distrubution in the Wasserstein sense. It is important

Figure 3: Plot of robust OT estimate for different values of $\rho$

Table 1: **Analyzing mode drop:** Training robust GAN on imbalanced CelebA. Each column denotes an experiment where GAN models are trained on input dataset having the respective fraction of males as given in row 1. Rows 2 and 3 denote the fraction of males in the generated dataset obtained by training Vanilla and Robust GAN respectively. We observe that images of males are generated even when the fraction of males in the input dataset is as low as 2%.

| Model | Fraction of males (in %) | | | |
|---|---|---|---|---|
| Input dataset | 2.00 | 5.00 | 10.00 | 20.00 |
| Vanilla GAN | 5.23 | 7.42 | 12.16 | 21.29 |
| Robust GAN | 4.84 | 7.80 | 10.12 | 21.51 |

that these dropped samples do not correspond to rare modes in the dataset, or have a mechanism to identify such mode dropping when it happens. In Table 1 of main paper, we observe no drop in FID scores on clean CIFAR-10 dataset, which suggests that no mode drop has occured. To further understand if biases in the dataset are exacerbated, we train our robust Wasserstein GAN model on CelebA dataset with varying male:female ratio. We then measure the male:female ratio of the generated distribution obtained from the trained GAN (using an attribute classifier). Table 1 shows the results. We observe that even when fraction of males in the input dataset are as low as 2%, images of males are generated in robust GAN model. This indicates that rare modes are not dropped. Instead, our robust GAN model drops samples having large noise.

**Identifying mode drop:** In some cases, as pointed by the reviewer, rare modes can potentially be dropped. In this case, we can use the weights estimated by our weight network $\mathbf{W}(x)$ to visualize which modes are dropped (Fig 3 and 4 of the main paper). Samples with low weights are the ones that are dropped. We can use these weights in a boosting framework to train a mixture model to generate balanced datasets. This is a topic of future research.

## 5   Properties of Robust OT

For a measure to be a distance metric, it has to satisfy four propeties of non-negativity, identity, symmetry and triangle inequality.

**Non-negativity**   Robust Wasserstein measure $\mathcal{W}^{rob}_{\rho_1,\rho_2}(\mathbb{P}_X, \mathbb{P}_Y)$ is non-negative. By definition,

$$\mathcal{W}^{rob}_{\rho_1,\rho_2}(\mathbb{P}_X, \mathbb{P}_Y) := \min_{\mathbb{P}_{\tilde{X}}, \mathbb{P}_{\tilde{Y}} \in Prob(\mathcal{X})} \min_{\pi \in \Pi(\mathbb{P}_{\tilde{X}}, \mathbb{P}_{\tilde{Y}})} \int \int c(x,y)\pi(x,y)dxdy$$

$$\text{s.t. } \mathcal{D}_f(\mathbb{P}_{\tilde{X}}||\mathbb{P}_X) \leq \rho_1, \ \mathcal{D}_f(\mathbb{P}_{\tilde{Y}}||\mathbb{P}_Y) \leq \rho_2.$$

Since the cost function $c(.)$ and the transportation map $\pi(.)$ are non-negative, the robust OT measure is non-negative.

**Identity:**   Robust Wasserstein measure satisfies identity. In other words, $\mathcal{W}^{rob}_{\rho_1,\rho_2}(\mathbb{P}_X, \mathbb{P}_X) = 0$. To prove this, consider the following solution: $\pi(x,y) = 1$ if $x = y$, and $\pi(x,y) = 0$ otherwise,

$\mathbb{P}_{\tilde{X}} = \mathbb{P}_X$ and $\mathbb{P}_{\tilde{Y}} = \mathbb{P}_Y$. Clearly, this is a feasible solution. Also, under this solution, the OT cost is 0 as $c(x, x) = 0$. Since, robust OT is non-negative, this is the optimal solution.

**Symmetry:** In general, robust Wasserstein is not symmetric $\mathcal{W}_{\rho_1,\rho_2}^{rob}(\mathbb{P}_X, \mathbb{P}_Y) = 0 \neq \mathcal{W}_{\rho_1,\rho_2}^{rob}(\mathbb{P}_Y, \mathbb{P}_X) = 0$. This is because when $\rho_1 \neq \rho_2$, different $f$-divergence constraints are imposed on the two marginals leading to different solutions. However, $\mathcal{W}_{\rho,\rho}^{rob}(\mathbb{P}_X, \mathbb{P}_Y)$ is symmetric. This is because $\int \int c(x, y)\pi(x, y)dxdy = \int \int c(y, x)\pi(y, x)dydx$, and constraints on two marginals are the same for both optimization problems $\mathcal{W}_{\rho,\rho}^{rob}(\mathbb{P}_X, \mathbb{P}_Y)$ and $\mathcal{W}_{\rho,\rho}^{rob}(\mathbb{P}_Y, \mathbb{P}_X)$.

**Triangle inequality:** $\mathcal{W}_{\rho_1,\rho_2}^{rob}(\mathbb{P}_X, \mathbb{P}_Y)$ does not satisfy triangle inequality.

# 6 Experiments

## 6.1 Generative modeling

The formulation of robust Wasserstein GAN is discussed in Section 5 of the main paper. The idea is to modulate the discriminator loss using a weight network $\mathbf{W}(.)$. The output of the $\mathbf{W}(.)$ has a ReLU transform to make it non-negative, and for each batch, the weights are normalized to satisfy $\mathbb{E}_{\mathbf{x} \in \mathbb{P}_X}[\mathbf{W}(\mathbf{x})] = 1$. Then, the objective of robust Wasserstein GAN can be written as

**Robust Wasserstein GAN:**

$$\min_{\mathbf{W}, \mathbf{G}} \max_{\mathbf{D} \in Lip-1} \mathbb{E}_{\mathbf{x} \sim \mathbb{P}_X}[\mathbf{W}(\mathbf{x})\mathbf{D}(\mathbf{x})] - \mathbb{E}_{\mathbf{z}}[\mathbf{D}(\mathbf{G}(\mathbf{z}))] + \lambda \max\left(\mathbb{E}_{\mathbf{x} \sim \mathbb{P}_X}[(\mathbf{W}(\mathbf{x}) - 1)^2] - 2\rho, 0\right) \tag{11}$$

We can similarly extend the robust GAN formulations to other GAN variants such as spectral normalization GAN or non-saturating GAN as follows.

**Robust Non-saturating GAN:**

$$\min_{\mathbf{W}, \mathbf{G}} \max_{\mathbf{D}} \mathbb{E}_{\mathbf{x} \sim \mathbb{P}_X}[\mathbf{W}(\mathbf{x})\log(\mathbf{D}(\mathbf{x}))] + \mathbb{E}_{\mathbf{z}}[\log(1 - \mathbf{D}(\mathbf{G}(\mathbf{z})))] + \lambda \max\left(\mathbb{E}_{\mathbf{x} \sim \mathbb{P}_X}[(\mathbf{W}(\mathbf{x}) - 1)^2] - 2\rho, 0\right) \tag{12}$$

**Robust Spectral Normalization GAN:**

$$\min_{\mathbf{W}, \mathbf{G}} \max_{\mathbf{D}} \mathbb{E}_{\mathbf{x} \sim \mathbb{P}_X}[\mathbf{W}(\mathbf{x})\max(1 - \mathbf{D}(\mathbf{x}), 0)] + \mathbb{E}_{\mathbf{z}}[\max(1 + \mathbf{D}(\mathbf{G}(\mathbf{z})), 0)]$$
$$+ \lambda \max\left(\mathbb{E}_{\mathbf{x} \sim \mathbb{P}_X}[(\mathbf{W}(\mathbf{x}) - 1)^2] - 2\rho, 0\right) \tag{13}$$

In what follows, we provide more details on the experimental settings used in main paper.

### 6.1.1 Dataset with outliers

In Section 5.1.1 of the main paper, we show experiments on CIFAR-10 dataset artificially corrupted with MNIST and uniform noise outliers. Each experiment is conducted with a different outlier fraction $\gamma$. For a given $\gamma$, the outlier-corrupted dataset is constructed so that outliers occupy $\gamma$ fraction of samples. In all experiments, the total size of the dataset (including outliers) was maintained as 50000. All models were trained using robust Wasserstein loss given in Eq. (11). Experiments were performed on two architectures - DCGAN and Resnet.

For quantitative evaluation, FID scores with respect to clean CIFAR-10 dataset are reported. That is, let $m_c$ and $C_c$ denote the mean and covariance matrices of Inception network features obtained from clean CIFAR-10 dataset (CIFAR-10 without outliers), and $m$ and $C$ denote the mean and covariance of Inception network features obtained from generated samples. Then, the FID score is computed as

$$FID = \|m - m_c\|_2^2 + Tr\left(C + C_c - 2(CC_c)^{1/2}\right).$$

To compute the FID score, the mean and covariance matrices were computed over 50000 samples.

On sketch domain on DomainNet dataset, robust spectral normalization GAN (Eq. (13)) was trained. Resnet architectures were used for both discriminator and generator.

**Algorithm 1** Robust Wasserstein GAN training algorithm

---

**Require:** $N_{iter}$: Number of training iterations, $N_{critic}$: Number of critic iterations, $N_{batch}$: Batch size, $N_{weight}$: Number of weight update iterations
1: **for** $t$ in $1 : N_{iter}$ **do**
2:     Sample a batch of real samples $\{\mathbf{x}_i\}_{i=1}^{N_{batch}} \sim \mathcal{D}$
3:     Sample a batch of noise vectors $\{\mathbf{z}_i\}_{i=1}^{N_{batch}} \sim \mathcal{D}$
4:     Normalize weight vectors as $\mathbf{W}(\mathbf{x}_i) \leftarrow \mathbf{W}(\mathbf{x}_i) / \sum_{i=1}^{N_{batch}} \mathbf{W}(\mathbf{x}_i)$
5:     Obtain GAN loss as

$$\mathcal{L}_{GAN} = \frac{1}{N_{batch}} \sum_i \mathbf{W}(\mathbf{x}_i)\mathbf{D}(\mathbf{x}_i) - \frac{1}{N_{batch}} \sum_i \mathbf{D}(\mathbf{G}(\mathbf{z}_i))$$

$$+ \lambda \max\left(\frac{1}{N_{batch}} \sum_i [(\mathbf{W}(\mathbf{x}_i) - 1)^2] - 2\rho, 0\right) + \lambda_{GP} \frac{1}{N_{batch}} \sum_i (\|\mathbf{D}(\hat{\mathbf{x}}_i)\| - 1)^2$$

6:     Update discriminator as $\mathbf{D} \leftarrow \mathbf{D} + \eta_D \nabla_{\mathbf{D}} \mathcal{L}_{GAN}$
7:     **if** $t \% N_{weight} == 0$ **then**
8:         Update weight network as $\mathbf{W} \leftarrow \mathbf{W} - \eta_w \nabla_{\mathbf{W}} \mathcal{L}_{GAN}$
9:     **end if**
10:    **if** $t \% N_{critic} == 0$ **then**
11:        Update generator as $\mathbf{G} \leftarrow \mathbf{G} - \eta_g \nabla_{\mathbf{G}} \mathcal{L}_{GAN}$
12:    **end if**
13: **end for**

---

Table 2: Architectures and hyper-parameters: Resnet model

| Generator | Discriminator |
|---|---|
| $z \in \mathbb{R}^{128} \sim \mathcal{N}(0, 1)$ | Input $\mathbf{x} \in \mathbb{R}^{32 \times 32 \times 3}$ |
| Dense, $4 \times 4 \times 128$ | ResBlock down 128 |
| ResBlock up 128 | ResBlock down 128 |
| ResBlock up 128 | ResBlock 128 |
| ResBlock up 128 | ResBlock 128 |
| BN, ReLU, Conv $3 \times 3$ | ReLU, Global sum pooling |
| Tanh | Dense $\rightarrow 1$ |

| Hyperparameters | |
|---|---|
| Generator learning rate | 0.0002 |
| Discriminator learning rate | 0.0002 |
| Weight net learning rate | 0.0002 |
| Generator optimizer | Adam, Betas $(0.0, 0.999)$ |
| Discriminator optimizer | Adam, Betas $(0.0, 0.999)$ |
| Weight net optimizer | Adam, Betas $(0.0, 0.999)$ |
| Number of critic iterations | 5 |
| Weight update iterations | 5 |
| Gradient penalty | 10 |
| Batch size | 128 |

Figure 4: Visualizing samples generated on CIFAR-10 dataset corrupted with MNIST outliers.

Figure 5: Visualizing samples generated on CIFAR-10 dataset corrupted with uniform noise outliers.

### 6.1.2 Clean datasets

In Section 5.1.2 of the main paper, robust Wasserstein GANs were trained on clean datasets (datasets with no outlier noise). Expeirments were performed on CIFAR-10 and CIFAR-100 datasets. On CIFAR-10, unconditional model was trained, whereas on CIFAR-100, conditional GAN using conditional batch normalization (similar to [3]) was trained.

### 6.1.3 Architectures and Hyper-parameters

Models and hyperparameters used for DCGAN and Resnet models are provided in Tables 2 and 3. For both models, the architecture used for the weight network is provided in Table 4.

Table 3: Architectures and hyper-parameters: DCGAN model

| Generator | Discriminator |
|---|---|
| $z \in \mathbb{R}^{128} \sim \mathcal{N}(0,1)$ | Input $\mathbf{x} \in \mathbb{R}^{32 \times 32 \times 3}$ |
| Dense, $4 \times 4 \times 512$ + BN + ReLU | Conv $3 \times 3$, str $1$, $(64)$ + LReLU |
| ConvTranspose $4 \times 4$, str $2$, $(256)$ + BN + ReLU | Conv $4 \times 4$, str $2$, $(128)$ + BN + LReLU |
| ConvTranspose $4 \times 4$, str $2$, $(128)$ + BN + ReLU | Conv $4 \times 4$, str $2$, $(256)$ + BN + LReLU |
| ConvTranspose $4 \times 4$, str $2$, $(64)$ + BN + ReLU | Conv $4 \times 4$, str $2$, $(512)$ + BN + LReLU |
| Conv $3 \times 3$, str $1$, $(3)$ + TanH | Conv $4 \times 4$, str $1$, $(1)$ |
| Hyperparameters | |
| Generator learning rate | 0.0001 |
| Discriminator learning rate | 0.0001 |
| Weight net learning rate | 0.0001 |
| Generator optimizer | Adam, Betas $(0.5, 0.9)$ |
| Discriminator optimizer | Adam, Betas $(0.5, 0.9)$ |
| Weight net optimizer | Adam, Betas $(0.5, 0.9)$ |
| Number of critic iterations | 5 |
| Weight update iterations | 5 |
| Gradient penalty | 10 |
| Batch size | 128 |

Table 4: Architectures of weight network

| Weight network |
|---|
| Input $\mathbf{x} \in \mathbb{R}^{32 \times 32 \times 3}$ |
| Conv $3 \times 3$, str $1$, $(64)$ + ReLU + Maxpool$(2 \times 2)$ |
| Conv $3 \times 3$, str $1$, $(128)$ + ReLU + Maxpool$(2 \times 2)$ |
| Conv $3 \times 3$, str $1$, $(256)$ + ReLU + Maxpool$(2 \times 2)$ |
| Conv $4 \times 4$, str $1$, $(1)$ |

## 6.2 Domain Adaptation

In all domain adaptation experiments, we update the weight vectors using discrete optimzation version discussed in Section 4.3.1 of the main paper. For all models, an entropy regularization on target logits is used. A complete algorithm is provided in Alg. 2. We evaluate our approach on VISDA-17 dataset using Resnet-18, 50 and 101 architectures as discussed in Section 5.2 of the main paper.

**Baselines:** Two standard baselines for the domain adaptation problem include source only and Adversarial alignment. In the source only model, the feature network $\mathbf{F}$ and the classifier $\mathbf{C}$ are trained only on the labeled source dataset. In the adversarial alignment, domain discrepancy between source and target feature distributions is minimized in addition to the source classification loss. This is essentially an unweighted version of our adaptation objective.

**Architectures:** In our experiments, the feature network $\mathbf{F}$ is implemented using Resnet architectures (Resnet-18, 50 and 101). Following the usual transfer learning paradigm, the last linear layer is removed in the Resnet network. The feature network is intialized with weights pretrained on Imagenet. The classifier network $\mathbf{C}$ is a linear layer mapping the features to probability vector with dimension equal to the number of classes. The classifier network is trained from scratch. The discriminator is realized as a 3-layer MLP with ReLU non-linearities and hidden layer dimension 256. Spectral normalization is used on the discriminator network. The hyper-parameters used in all experiments are described in Table 5.

**Weight visualization:** We visualize the weights learnt by our domain adaptation module by plotting the histogram of weights in Figure 7. Additionally, target samples sorted by weights are shown in Figure 6.

---

**Algorithm 2** Domain adaptation training algorithm

---

**Require:** $N_{iter}$: Number of training iterations, $N_{critic}$: Number of critic iterations, $N_{batch}$: Batch size, $N_{weight}$: Number of weight update iterations
1: Intialize weight bank $\mathbf{w}_b = [w_1, w_2, \ldots w_{N_t}]$ , where each $w_i$ corresponds to weight of target $\mathbf{x}_i^t$
2: **for** $t$ in $1 : N_{iter}$ **do**
3:     Sample a batch of labeled source images $\{\mathbf{x}_i^s, y_i^s\}_{i=1}^{N_{batch}} \sim \mathcal{D}_s$ and unlabeled target images $\{\mathbf{x}_i^t\}_{i=1}^{N_{batch}} \sim \mathcal{D}_t$
4:     Obtain the weight vectors $w_i^t$ corresponding to the target samples $\mathbf{x}_i^t$ from the weight bank $\mathbf{w}_b$
5:     Obtain discriminator loss as

$$\mathcal{L}_{disc} = \frac{1}{N_{batch}} \sum_i L_{BCE}(\mathbf{D}(\mathbf{F}(\mathbf{x}_i^s)), 0) + \frac{1}{N_{batch}} \sum_i w_i^t L_{BCE}(\mathbf{D}(\mathbf{F}(\mathbf{x}_i^t)), 1)$$

6:     Obtain source label prediction loss as

$$\mathcal{L}_{cls} = \frac{1}{N_{batch}} \sum_i L_{CE}(\mathbf{C}(\mathbf{F}(\mathbf{x}_i^s)), y_i^s)$$

7:     Update discriminator $\mathbf{D} \leftarrow \mathbf{D} - \eta_d \nabla_{\mathbf{D}} \mathcal{L}_{disc}$
8:     Update feature network and classifier $\mathbf{F} \leftarrow \mathbf{F} - \eta_f \nabla_{\mathbf{F}} \mathcal{L}_{cls}$, $\mathbf{C} \leftarrow \mathbf{C} - \eta_c \nabla_{\mathbf{C}} \mathcal{L}_{cls}$
9:     **if** $t \% N_{weight} == 0$ **then**
10:         Update weight bank $\mathbf{w}_b$ using Algorithm 3
11:     **end if**
12:     **if** $t \% N_{critic} == 0$ **then**
13:         Update feature network as $\mathbf{F} \leftarrow \mathbf{F} + \eta_f \nabla_{\mathbf{F}} \mathcal{L}_{disc}$
14:     **end if**
15: **end for**

---

---

**Algorithm 3** Algorithm for updating weights

---

1: Form the discriminator vector $\mathbf{d} = [\mathbf{D}(\mathbf{x}_1^t), \mathbf{D}(\mathbf{x}_2^t), \ldots \mathbf{D}(\mathbf{x}_{N_t}^t)]$
2: Obtain $\mathbf{w}_b$ as the solution of the following second-order cone program

$$\min_{\mathbf{w}} (\mathbf{w})^t \mathbf{d}$$
$$\text{s.t } \|\mathbf{w} - 1\|_2 \leq \sqrt{2\rho_1 N_t}$$
$$\mathbf{w} \geq 0, \ (\mathbf{w})^t \mathbf{1} = N_t$$

3: Return $\mathbf{w}_b$

---

Table 5: Hyper-parameters for domain adaptation experiments

| Hyperparameters | |
|---|---|
| **F** network learning rate | 0.0005 |
| **C** network learning rate | 0.0005 |
| **D** network learning rate | 0.0001 |
| **F** network optimizer | SGD with Inv LR scheduler |
| **C** network learning rate | SGD with Inv LR scheduler |
| **D** network learning rate | Adam betas $(0.9, 0.999)$ |
| Number of critic iterations | 5 |
| Batch size | 28 |
| Entropy weight | 0.25 |
| Weight update iterations | 500 |
| Outlier fraction $\rho$ | 0.2 |

Figure 6: Visualizing target domain samples sorted by learnt weights

Figure 7: Visualizing histograms of weights learnt in robust domain alignment