[Reviews · NeurIPS 2020]

Review 1

Summary and Contributions: The paper presents a robust optimal transport formulation to handle outliers. As the part of learning objective, it learns weights for each sample such that the outlier samples get lower weights. The method's application is shown on GAN and UDA.

Strengths: - Novel method and achieves good experimental results. - Well written paper; easy-to-follow, clearly positions the proposed method with the existing approaches. - Overall satisfactory evaluation.

Weaknesses: There are two concerns I have with this work. - The method could lead to an increase or exacerbation of the biases present in the dataset. Consider training face generator, the face datasets like CelebA-HQ and FFHQ have some strong biases like largely Caucasian faces, very few blond males etc. Now training a GAN on these datasets with the proposed method, the GAN might completely or almost drop African faces or blond-male faces because they are very few in the dataset and further weighted low by robust OT method. On the other hand it might increase the presence of Caucasian-smiling-female faces (dominant attributes) due to higher presence and larger weights. - Possibility of mode drop. What prevents this method from mode drop when the mode is difficult to model? For example, if you want to train a generator with 60-70% MNIST and 30-40% SVHN, the SVHN images are difficult compared MNIST images. I think there is a risk that the generator trained with robust OT method might completely drop the difficult images, which are definitely not outliers given the 30-40% presence. My overall suggestion is to have an analysis on what is the impact of the method on the distribution, which is to observe the percentage of each class in the generated data or class wise performance in the case of UDA. If there are some drawbacks in these cases then, it should be presented. Another point, related to the above points, what happens if you flatten the weights or make it uniform except for the outliers. It could be a trick to circumvent issues of biases or mode drop.

Correctness: The empirical methodology and claims seems correct.

Clarity: The paper is well written.

Relation to Prior Work: The related work is well discussed and the proposed work is also well placed through this discussion.

Reproducibility: Yes

Additional Feedback: ## Post-rebuttal ## I find the rebuttal convincing for the issues raised in the reviews. My concerns about inducing biases and mode drop are well addressed with experimental results. Therefore, I increase my rating to 7.


Review 2

Summary and Contributions: In the paper, the authors proposed a communicationally-efficient dual form of the robust optimal transport (OT) optimization that is useful for large-scale deep learning applications. In particular, the main idea of the paper is motivated from a robust formulation of OT with unbalanced marginal constraints, which is the popular unbalanced optimal transport (UOT) in the literature. However, training the neural networks with the dual form of UOT is challenging as it involves maximizing over two discriminator functions with some constraints. Therefore, the authors proposed to constrain the transportation plan in the UOT problem to have its marginal constraints, which they refer to as relaxed distribution, lying in the probability space and the f-divergences between the target distributions and the relaxed distributions to be bounded by some given constants. They demonstrated that their formulation can work efficiently with applications of GAN and domain adaptation.

Strengths: In my opinion, the robust optimal transport proposed by the authors is quite interesting. The careful experiments with GAN and domain adaptation demonstrate the effectiveness of the framework for dealing with outliers. The theoretical results in Theorem 1 and Theorem 2 are reasonable. I checked the proofs of these results in the Supplementary material and they all seem correct to me.

Weaknesses: There are a few concerns with the paper that the authors should address: (1) Regarding the formulation of robust optimal transport in equation (5), when the probability measures $P_{X}$ and $P_{Y}$ are discrete, the objective function with $\pi$ is a linear programming problem. Solving directly the linear programming problem can be expensive as the standard interior point methods have complexity of order $n^3$ where $n$ is the maximum number of supports of $P_{X}$ and $P_{Y}$. For this reason, we can utilize the entropic regularized approach to equation (5). Eventually, it leads to a strongly convex objective function as follows: \min_{P_{\tilde{X}}, P_{\tilde{Y}}} \min_{\pi} \pi_{ij} C_{ij} - \eta H(\pi), s.t. D_{f}(P_{\tilde{X}}|| P_{X}) \leq \rho_{1}, D_{f}(P_{\tilde{Y}}|| P_{Y}) \leq \rho_{2}. Here, $H(\pi)$ stands for the entropy of $\pi$. The above objective function is equivalent to solving the following optimization problem: \min_{P_{\tilde{X}}, P_{\tilde{Y}}} \min_{\pi} \pi_{ij} C_{ij} + \tau_{1} D_{f}(P_{\tilde{X}}|| P_{X}) \leq + \tau_{2} D_{f}(P_{\tilde{Y}}|| P_{Y}) \leq \rho_{2} - \eta H(\pi), where $\tau_{1}, \tau_{2}, \eta > 0$ are some given positive numbers. The objective function is strongly convex with respect to $\pi$. When $D_{f}$ is KL-divergence, the dual form of that objective function has very nice form (see the reference [1]). We essentially can use Sinkhorn algorithm to solve it. The Sinkhorn framework is also differentiable; therefore, it can be used for deep learning framework. Now, my question is: how efficient the current proposed objective functions (7) and (8) comparing to the Sinkhorn framework when we utilize the entropic regularized framework? It will be great if the authors can provide some initial experiments on the performance of Sinkhorn algorithm for their problem when $P_{X}$ and $P_{Y}$ are discrete. (2) I am in fact confused about the objective function (8). How is it easier to implement than the objective function (7)? It will be helpful if the author can provide an explicit algorithm for solving (8) in the main text or in the supplementary material. Note that, the complexity of the standard algorithms for solving both (7) and (8) are at least at the order of $n \times m$ (up to some accuracy level) where $n$ and $m$ are respectively the number of supports of $P_{X}^{(m)}$ and $P_{Y}^{(n)}$. When both $n$ and $m$ are large, these standard algorithms are not scalable. Therefore, we need to devise careful optimization algorithms in order to deal with the large-scale of $n \times m$. (3) Assume that we have $n$ i.i.d. data from the corrupted distribution $P_{X} = (1 - \gamma) P_{X}^{c} + \gamma P_{X}^{a}$. Denote $P_{n}$ as the empirical measure of these data. What can we say about the gap between $P_{n}$ and $P_{X}$ under the robust optimal transport based on $n$? Can we guarantee that the gap goes to 0 as the sample size goes to infinity? (4) Typos: --- In Theorem 1, the function k in the maximum in equation (1) should be function D. Please also fix the proof of Theorem 1 in the Appendix accordingly as the notation are not consistent with that in the main paper. --- Line 137: $P_{Y}^{(m)}$ should be changed to $P_{Y}^{(n)}$. --- Line 140: $w^{x} \in \Delta^{m}$ should be changed to $w_{x} \in \Delta^{m}$.

Correctness: I checked most of the claims, the theories, and the experiments. They seem correct to me.

Clarity: The paper is quite well-written. There are a few typos in the main paper for which I already listed. The authors should try to proofread the paper more to make sure that it is typo-free.

Relation to Prior Work: The paper does not cite several related recent work on unbalanced optimal transport. A few examples of these work are: [1] K. Pham, K. Le, N. Ho, T. Pham, H. Bui. On Unbalanced Optimal Transport: An Analysis of Sinkhorn Algorithm. ICML, 2020. [2] T. Séjourné, J. Feydy, F. Vialard, A. Trouvé, G. Peyré. Sinkhorn Divergences for Unbalanced Optimal Transport. arXiv preprint arXiv: 1910.12958, 2019.

Reproducibility: Yes

Additional Feedback:


Review 3

Summary and Contributions: The authors are motivated by three factors: (A) the increasing use of OT distances in objective functions for generative modeling (specifically GANs), and domain translation; (B) the sensitivity of the OT distances to outliers; and (C) the difficulty employing existing robust OT solvers in deep learning due to their instability. The authors introduce a novel, and efficient robust OT measure (a variant of the wasserstein distance). Their formulation can be implemented as learning a re-weighting of the training data, with bounded divergence from the uniform distribution. They demonstrate the utility of their method for training GANs where the training data contains outliers, and for improving accuracy in domain adaptaion. In addition they prove that their proposed robust OT measure is upper bounded by a constant factor relative to the true OT distance when outliers are removed from the distribution.

Strengths: + The authors experiments qualitatively and quantitatively demonstrate that their measure improves the sample quality of GANs when the dataset is intentionally corrupted with outliers (CIFAR+MNIST) or when the dataset naturally contains outliers (DomainNet Sketch). They also show that it does not greatly affect sample quality when the dataset is clean (CIFAR) + The sample weights learned by their method gives a simple way of interpreting which portions of the distribution were ignored during training + The proposed robust adversarial measure dramatically improves of the adversarial baseline when used for domain adaptation. Result improve further to be on par with competing methods when entropy regularization is applied to the the target logits. + The authors derive two versions of their measure, one suitable to discrete distributions, and the other suitable to continuous distributions (or large datasets), and describe the practical details of using both. + Clarity: the authors clearly explain their proposed measure by comparing and constrasting it with unbalanced OT, and the Kantrovich-Rubeninstein duality.

Weaknesses: - The authors do not explore the effect of their formulation on more recent GAN architectures, or high resolution datasets.

Correctness: The empirical claims of the author are well supported in the main text and supplement, and the proofs of the author's theoretical claims in the supplement seem correct.

Clarity: Yes, the paper is well-written and easy to follow.

Relation to Prior Work: Yes, this is discussed throughout the paper, despite the lack of a centralized related works section.

Reproducibility: Yes

Additional Feedback: ### Post-Rebuttal Comments ### After reading the rebuttal I agree that the authors evaluate their method on generally accepted benchmarks for the field, and this addresses my concern. I am keeping my previous score. ###


Review 4

Summary and Contributions: This paper proposes to optimize unbalanced OT with a computationally effective dual form. Two solvers for the dual problem are developed and shows high stability in large-scale applications. The theoretical analysis also demonstrate the robust OT can handle outliers. Both the empirical end theorical results show the effectiveness of the proposed method.

Strengths: +The paper is well-written. The motivation of the proposed method is well clarified in the Introduction. The backgrounds in Section 2 also help readers understand. The method is described clearly and easy to follow. +The theorical analysis is solid. Theorem 2 in Section 3.3 shows that the robust OT can be upper bounded. The proposed method is demonstrated to be effective from a theoretical perspective. +The experiment results in multiple datasets show the superiority of the proposed method.

Weaknesses: -The proposed methods is both used in generative models and domain adaptation. Actually, when applying a method on the two areas (i,e., generative models and domain adaptation), special design may be needed. Could the authors compare the difference more detailly between the way of applying the proposed method on the two areas? -For domain adaptation experiments, the proposed method is compared with the existing methods on VISDA-17. Experiments on more datasets (e.g., Office-31) are needed.

Correctness: Yes. Yes.

Clarity: This paper is well written and easy to read.

Relation to Prior Work: Yes.

Reproducibility: Yes

Additional Feedback:

[Author Response · NeurIPS 2020]

We thank all the reviewers for their valuable feedback and the positive evaluations of our paper. Below, we address the
comments of each reviewer individually.

**[R1] Biases and Mode Drop:** The model of outliers we assume in our paper is that of large noise (Section 3). Hence,
our model drops samples that are far from the true data distrubution in the Wasserstein sense. As pointed by the reviewer,
it is important not to drop rare modes, or have a mechanism to identify mode dropping when it happens. As shown in
Table 1 of the main paper, there is no drop in FID scores on clean CIFAR-10 dataset, which suggests that no mode drop
has occured. To further understand the effect of biases, we train our robust Wasserstein GAN model on CelebA dataset
with varying female:male ratio. We then measure the female:male ratio of the generated distribution obtained from the
trained GAN (using an attribute classifier). Table 1 shows the results. We observe that even when fraction of males are
as low as $2\%$ in the input dataset, they are generated with a similar ratio in GANs. We will add these results to the paper.

Table 1: **Analyzing mode drop:** Training robust GAN on imbalanced CelebA. In each column, we report $\%$ of males generated by a GAN trained on the respective input dataset.

| % Males - Input dataset | 2 % | 5 % | 10 % | 20 % |
|---|---|---|---|---|
| % Males - Vanilla GAN | 5.23 % | 7.42 % | 12.16 % | 21.29 % |
| % Males - Robust GAN | 4.84 % | 7.80 % | 10.12 % | 21.51 % |

Table 2: **Partial Domain Adaptation Results:** Office-31 dataset. Accuracy in % is reported

| Setting | W->A | D->W | D->A | A->W | W->D | A->D | Avg |
|---|---|---|---|---|---|---|---|
| Source only | 71.7 | 94.5 | 73.1 | 54.5 | 94.2 | 65.6 | 75.6 |
| Robust OT | 92.8 | 95.7 | 93.4 | 87.9 | 97.2 | 85.5 | 92.1 |

**[R1] Identifying mode drop:** In some cases, as pointed by the reviewer, rare modes can potentially be dropped. In
this case, we can use the weights estimated by our weight network $\mathbf{W}(x)$ to visualize which modes are dropped (Fig
3 and 4 of the main paper). Samples with low weights are the ones that are dropped. We can use these weights in a
boosting framework to train a mixture model to generate balanced datasets. We will add this discussion to the paper.

**[R2] Comparison with Sinkhorn Iterations:** We thank the reviewer for pointing us to the reference. The focus of
our paper is to present relaxations to make unbalanced OT suited for deep learning applications. Hence, to compare our
algorithm with [1], we trained a Resnet WGAN model on CIFAR-10 dataset as follows: We sample a batch of real and
generated samples and consider these as empirical distribution of size $n$. For these distributions, we solve for the dual
vectors $u$ and $v$ for $K$ iterations using the Sinkhorn algorithm in [1]. Then, we use these solutions in the unbalanced OT
dual objective and optimize for the generator. We then iterate between these two steps. We observed that the GAN
model trained using this procedure produced poor generations (extremely blurred samples with mode collapse). Similar
issues on using entropic GAN objectives for deep learning have been reported in [2].
[1] "On Unbalanced Optimal Transport: An Analysis of Sinkhorn Algorithm", ICML 2020.
[2] "Entropic GANs meet VAEs: A Statistical Approach to Compute Sample Likelihoods in GANs", ICML 2019.

**[R2] Optimization Objectives:** Objective (8) is easier to implement than (7) for neural networks because in (8),
weights are obtained using a neural net. This makes the entire network end-to-end differentiable and the optimization
can be approximately solved using SGD. In objective (7), however, we need to alternate between SGD steps for training
GAN and a second-order cone optimization for estimating weights (which can be expensive for large datasets). Explicit
algorithm for solving (7) and (8) (for GANs and DA) are provided in Alg. 1, 2 and 3 of Supplementary material.

**[R2] Asymptotic convergence:** Yes, the gap between empirical robust Wasserstein measure $\mathcal{W}^{rob}(\mathbb{P}_X^{(n)}, \mathbb{P}_Y^{(n)})$ and true
robust Wasserstein $\mathcal{W}^{rob}(\mathbb{P}_X, \mathbb{P}_Y)$ goes to 0 as $n \to \infty$. The proof is very similar to the asymptotic convergence proof
of unbalanced optimal transport as provided in reference [13]. We will explain it in the paper.

**[R3] Recent architectures and high resolution datasets:** In the main paper, we have provided results on Resnet-based
GAN and Spectral Normalization GANs on various datasets including CIFAR ($32 \times 32$ resolution) and CelebA ($64 \times 64$
resolution), which are benchmark datasets for GANs. We did not add perform experiments on large-scale higher
resolution datasets like Imagenet due to the lack of computational resources.

**[R4] Special design choices:** Both GANs and domain adaptation experiments use distribution matching in their
objective: GANs minimize distributional distance between real and generated samples, while in domain adaptation,
we minimize distributional distance between source and target features. There are small changes in design choices,
such as discriminators operating in feature vs image space, architectures, etc. A detailed description of algorithm and
architectures are provided in supplementary material. Effectiveness on a wide range of problems shows the versatility
of our proposed approach. We will further add a discussion in the revised draft.

**[R4] Office-31 Experiments:** Upon your suggestion, we performed experiments on partial domain adaptation problem
on Office-31 dataset following the protocol provided in [3]. Results are obtained as shown in Table. 2. Our approach
achieves good performance improvement compared to baseline.
[3] Cao et al., "Partial Adversarial Domain Adaptation", ECCV 2018.

**[R1-4] Typos.** Typos will be fixed and citations will be added.

[Meta-Review · NeurIPS 2020]

In this paper, authors propose a robust Optimal transport method for generative model. The proposed approach is formulated as a min-max problem and the formulation is natural and makes sense. All reviewer's are happy with the proposed approach, so, I recommend the acceptance for this paper.